# The structural basis for dynamic DNA binding and bridging interactions which condense the bacterial centromere

Gemma LM Fisher[1†], César L Pastrana[2†], Victoria A Higman[3†], Alan Koh[4], James A Taylor[1‡], Annika Butterer[5§], Timothy Craggs[6], Frank Sobott[5,7,8], Heath Murray[4], Matthew P Crump[3], Fernando Moreno-Herrero[2], Mark S Dillingham[1]*

[1]DNA:protein Interactions Unit, School of Biochemistry, University of Bristol, Bristol, United Kingdom; [2]Department of Macromolecular Structures, Centro Nacional de Biotecnologia, Consejo Superior de Investigaciones Cientificas, Madrid, Spain; [3]School of Chemistry, University of Bristol, Bristol, United Kingdom; [4]Centre for Bacterial Cell Biology, Institute for Cell and Molecular Biosciences, Newcastle University, Newcastle, United Kingdom; [5]Biomolecular and Analytical Mass Spectrometry Group, Department of Chemistry, University of Antwerp, Antwerpen, Belgium; [6]Department of Chemistry, University of Sheffield, Sheffield, United Kingdom; [7]Astbury Centre for Structural Molecular Biology, University of Leeds, Leeds, United Kingdom; [8]School of Molecular and Cellular Biology, University of Leeds, Leeds, United Kingdom

*For correspondence:
mark.dillingham@bristol.ac.uk

[†]These authors contributed equally to this work

Present address: [‡]Laboratory of Molecular Biology, National Institute of Diabetes and Digestive and Kidney Diseases, National Institutes of Health, Bethesda, United States; [§]Institute of Pharmacy and Biochemistry, Johannes Gutenberg-Universität Mainz, Mainz, Germany

Competing interests: The authors declare that no competing interests exist.

**Abstract** The ParB protein forms DNA bridging interactions around *parS* to condense DNA and earmark the bacterial chromosome for segregation. The molecular mechanism underlying the formation of these ParB networks is unclear. We show here that while the central DNA binding domain is essential for anchoring at *parS*, this interaction is not required for DNA condensation. Structural analysis of the C-terminal domain reveals a dimer with a lysine-rich surface that binds DNA non-specifically and is essential for DNA condensation in vitro. Mutation of either the dimerisation or the DNA binding interface eliminates ParB-GFP foci formation in vivo. Moreover, the free C-terminal domain can rapidly decondense ParB networks independently of its ability to bind DNA. Our work reveals a dual role for the C-terminal domain of ParB as both a DNA binding and bridging interface, and highlights the dynamic nature of ParB networks in *Bacillus subtilis*.

DOI: https://doi.org/10.7554/eLife.28086.001

## Introduction

Bacterial chromosomes are actively segregated and condensed by the ParABS system and condensin (*Wang et al., 2014*; *Song and Loparo, 2015*). In *B. subtilis*, this machinery is physically targeted to the origin proximal region of the chromosome by eight palindromic DNA sequences called *parS* (consensus sequence 5′-TGTTNCACGTGAAACA-3′) to which the ParB (Spo0J) protein binds (*Breier and Grossman, 2007*; *Lin and Grossman, 1998*). These nucleoprotein complexes act as a positional marker of the origin and earmark this region for segregation in a manner somewhat analogous to eukaryotic centromeres and their binding partners.

ParB is an unusual DNA binding protein. In addition to sequence-specific interactions with the *parS* sequence, the protein also spreads extensively around the site for about 18 kbp (*Breier and Grossman, 2007*; *Murray et al., 2006*; *Lynch and Wang, 1995*). The mechanistic basis for this

behaviour is not well understood and a matter of active debate. Earlier models envisioned a lateral 1D spreading around *parS* to form a filament (*Murray et al., 2006*; *Rodionov et al., 1999*), princi-pally because spreading can be inhibited in a polar manner by 'roadblocks' placed to the side of *parS* sequences. However, ParB foci appear to contain fewer proteins than are necessary to form a filament, and single molecule analyses using direct imaging (*Graham et al., 2014*) and magnetic tweezers (*Taylor et al., 2015*) have shown that binding of DNA by ParB is accompanied by conden-sation. These 'networks' were inferred to be dynamic and poorly-ordered, consisting of several DNA loops between distally bound ParB molecules. In cells, they are presumably anchored at *parS* sites by sequence-specific interactions but must also contain many interactions with non-specific DNA (nsDNA), as well as self-association interactions that bridge ParB protomers to form DNA loops. Modelling suggests that a combination of 1D spreading and 3D bridging interactions can explain the condensation activity and recapitulate the polar effect of roadblocks on ParB spreading (*Broedersz et al., 2014*). Recently, single-molecule imaging of the F-plasmid SopB led to a broadly similar model, defining ParB networks as fluid structures that localise around *parS* using a 'nucleation and caging' mechanism (*Sanchez et al., 2015*). Despite these recent experiments converging on DNA bridging models to explain the ParB spreading phenomenon, the mechanism underpinning this behaviour remains unresolved. In particular, the relationship between these dynamic nucleopro-tein complexes and the molecular architecture of the ParB protein is unclear and is the subject of the work presented here.

Genomically-encoded ParB proteins comprise three distinct domains (*Figure 1A* and *Figure 1—figure supplement 1A,B and C*). Our understanding of their structure is limited to the N-terminal domain (NTD) which binds ParA (*Bouet and Funnell, 1999*; *Davey and Funnell, 1997*; *Davis et al., 1992*; *Radnedge et al., 1998*) and the central DNA binding domain (CDBD) which binds *parS* and possibly also nsDNA (*Leonard et al., 2004*; *Schumacher and Funnell, 2005*). A structure of *Thermus thermophilus* ParB lacking the C-terminal domain (CTD) revealed a compact dimer in which the helix-turn-helix (HtH) motifs were symmetrically arranged in a manner suitable for binding to the pal-indromic *parS* sequence (*Figure 1—figure supplement 1D*) (*Leonard et al., 2004*). Analysis of the

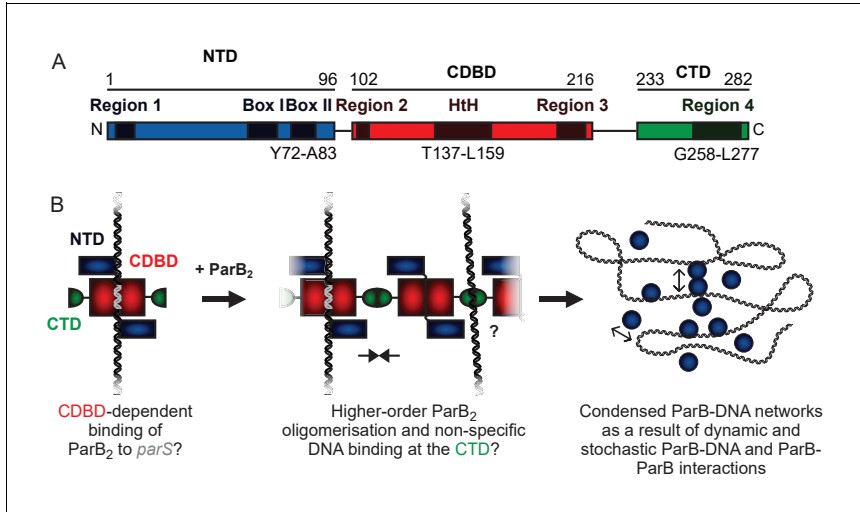

**Figure 1.** A hypothetical model for ParB-mediated condensation of the origin of replication region. (**A**) Domains and regions as identified in (*Bartosik et al., 2004*; *Kusiak et al., 2011*). (**B**) ParB is thought to be anchored at *parS* (grey) via the HtH motif found in the CDBD (red). ParB protomers self-associate via poorly defined interactions and also make non-specific contacts with DNA segments, leading to the formation of ParB networks. In this work we have investigated the potential role of the CTD (green) in mediating ParB oligomerisation and non-specific DNA binding.

DOI: https://doi.org/10.7554/eLife.28086.002

The following figure supplement is available for figure 1:

**Figure supplement 1.** Structural models for genomic ParB.
DOI: https://doi.org/10.7554/eLife.28086.003

CTD by analytical ultracentrifugation suggested that it also formed a dimer, and it was argued that this interface might promote spreading interactions. Recently, a structure of *Helicobacter pylori* ParB, in which the protein was also truncated by removal of the CTD, showed a strikingly different conformation, where the NTD had moved away from the CDBD domain to form a tetrameric self-association interface (*Figure 1—figure supplement 1E*) (*Chen et al., 2015*). The CDBD was bound to a *parS* half site, and it was argued that tetramerisation of the NTD could be responsible for bridging interactions between specific and nsDNA bound to the CDBD.

In previous work, we hypothesised that ParB contains a second DNA binding locus for nsDNA that functions independently of the HtH motif (*Figure 1B*) (*Taylor et al., 2015*). This idea was attractive to us for several reasons. Firstly, in a single DNA binding locus model, it is not straightforward to reconcile the strict localisation of ParB networks to just a few *parS* sites (and their surroundings) with the limited discrimination between specific and nsDNA binding that is observed in vitro (a <10 fold apparent difference in affinity) (*Taylor et al., 2015*; *Broedersz et al., 2014*). Secondly, although binding to *parS* protects the CDBD region from proteolysis, binding to nsDNA affords no such protection, implying that it interacts elsewhere on the protein (*Taylor et al., 2015*). Thirdly, the distantly-related ParB protein from plasmid P1 provides a precedent for a second DNA binding locus in a Type I centromere binding protein (albeit an additional *specific* DNA binding site), and highlights the CTD as the putative candidate region (*Schumacher et al., 2007*). However, the lack of any structural information for the CTD of a genomically-encoded ParB prevents a rigorous comparison of the systems because the primary structure similarity in this region is negligible.

In this work, we have probed the role of the CDBD and CTD of *B. subtilis* ParB using a combination of structural, biochemical, single molecule and in vivo approaches. We find that while the CDBD is responsible for specific recognition of *parS*, the CTD provides both a second nsDNA binding site and a self-association interface that is important for bridging interactions and DNA condensation.

## Results

### The R149 residue within the HtH motif is essential for specific binding to *parS*, but not required for non-specific binding and condensation

Genetic and structural analyses have suggested that residue R149 may be critically important for specific binding to *parS* at the HtH locus (*Graham et al., 2014*; *Chen et al., 2015*; *Autret et al., 2001*; *Gruber and Errington, 2009*). To probe the role of the HtH motif using biochemical techniques, we compared binding of *parS* by wild type ParB and ParB$^{R149G}$ using electrophoretic mobility shift assays containing $Mg^{2+}$ cations (TBM-EMSA). As reported previously, inclusion of divalent cations in both the gel composition and running buffer enables the clear differentiation of specific and nsDNA-binding activities of ParB (*Taylor et al., 2015*). As expected, binding of wild type ParB to *parS*-containing DNA produced a distinct band shift corresponding to the ParB$_2$-*parS* complex, as well as poorly migrating species at high [ParB] (*Figure 2A*). These latter complexes are also formed on DNA that does not contain *parS*, and are therefore indicative of ParB bound to nsDNA flanking the central *parS* sequence. ParB, and mutants thereof, were purified to homogeneity (*Figure 2—figure supplement 1A*). EMSA experiments with ParB$^{R149G}$ fail to produce the specific ParB$_2$-*parS* complex whereas the formation of nsDNA complexes is largely unaffected (*Figure 2A*). The retention of nsDNA binding activity in ParB$^{R149G}$ is further supported by data using gels lacking $Mg^{2+}$ ions (TBE-EMSA) (*Figure 2—figure supplement 1C*), as well as a solution-based protein-induced fluorescence enhancement (PIFE) assay (*Figure 2—figure supplement 1B*), in which an increase in Cy3 intensity reports ParB binding. For wild type ParB, the data were fitted to the Hill equation yielding an apparent $K_d$ of 361 ± 14 nM and Hill coefficient of 3.2 ± 0.3 in reasonable agreement with published data (*Lin and Grossman, 1998*; *Taylor et al., 2015*). ParB$^{R149G}$ produced a similar binding isotherm yielding a moderately weaker $K_d$ of 493 ± 18 nM. This apparent $K_d$ was not significantly altered when the Hill coefficient was not shared between datasets indicating the cooperativity of binding was not impaired in this ParB variant.

We next investigated the ability of ParB$^{R149G}$ to condense DNA tethers using magnetic tweezers (*Figure 2B*). We previously showed that wild type ParB mediates progressive condensation of DNA substrates which is reversible by both force and protein unbinding (*Taylor et al., 2015*). The condensed state is not highly ordered and its formation is not dependent upon *parS* sequences,

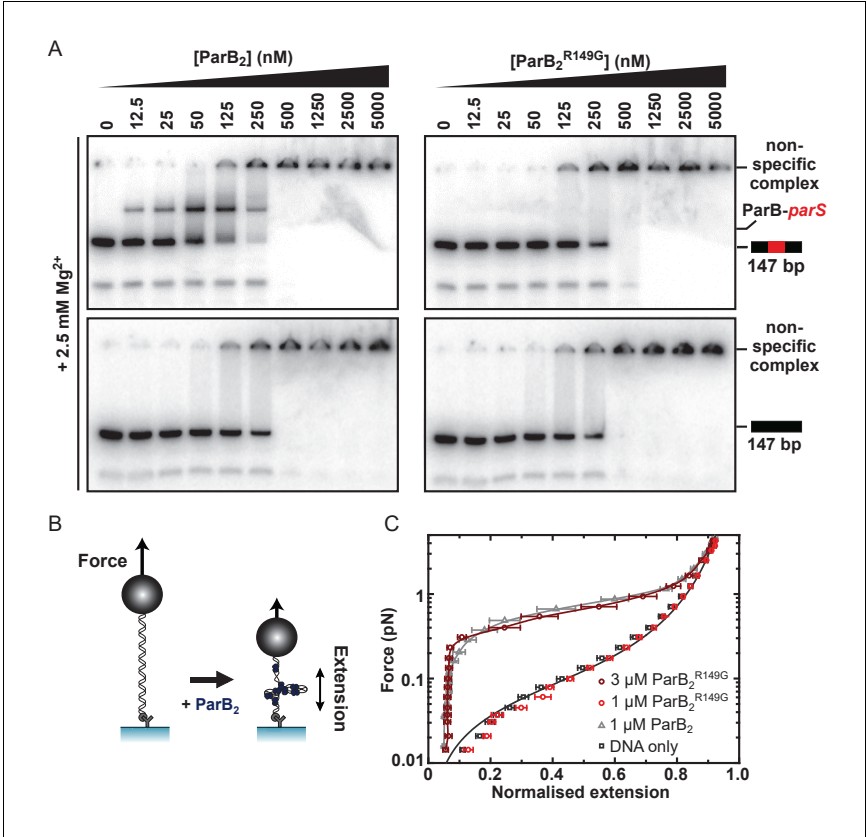

**Figure 2.** The R149 residue within the HtH motif is essential for specific binding to *parS*, but not required for non-specific binding and condensation. (**A**) Representative TBM-EMSAs for wild type ParB and ParB[R149G] monitoring binding of *parS*-containing or non-specific 147 bp dsDNA. (**B**) Schematic of the magnetic tweezer assay used to monitor ParB-dependent DNA condensation. (**C**) Mean force-extension curves for *parS*-containing DNA molecules in the presence of wild type ParB and ParB[R149G]. Non-condensed DNA data is fitted to the worm-like chain model. Solid lines in condensed data are guides for the eye. Errors are the standard error of the mean of measurements on different molecules (N ~15–35 molecules).

DOI: https://doi.org/10.7554/eLife.28086.004

The following figure supplement is available for figure 2:

**Figure supplement 1.** The R149 residue within the HtH motif is not required for non-specific binding and condensation.

DOI: https://doi.org/10.7554/eLife.28086.005

indicating that nsDNA binding is required for condensation. At a concentration sufficient for efficient condensation by wild type ParB (1 µM), ParB[R149G] did not fully condense DNA, although fluctuations of the DNA tether were consistent with minor condensation events that do not greatly affect the mean extension value measured (data not shown). However, at moderately elevated concentrations (3-fold), reversible condensation did occur and was qualitatively equivalent to wild type behaviour (*Figure 2C* and *Figure 2—figure supplement 1D*).

Together, these data show that mutation of the HtH motif effectively eliminates the ability of ParB to interact specifically with its cognate *parS* site, while nsDNA binding and condensation is relatively unaffected. This is consistent either with the R149G mutation exclusively affecting nucleobase-specific contacts in the ParB-*parS* complex, and/or with the idea that nsDNA binding may occur at a second DNA binding locus.

# The structure of *Bs*ParB CTD reveals a dimer with a putative DNA binding interface

We next used solution NMR to determine the structure of the CTD alone (see *Figure 3—source data 1* for structure validation and statistics and *Figure 3—figure supplement 1A* for an assigned $^1$H-$^{15}$N HSQC spectrum). The structure forms a well-defined dimer containing two α-helices and two β-strands per monomer in a α1-β1-β2-α2 arrangement (*Figure 3A and B*). The dimer interface is formed via an intermolecular β-sheet and two domain-swapped C-terminal helices. Although this protein fold is somewhat similar to that seen in the plasmid P1 and SopB ParB proteins (*Figure 3—figure supplement 1B,C and D*) (*Schumacher and Funnell, 2005*; *Schumacher et al., 2007*), there are also significant differences especially in the N-terminal region: the α1 helix in the chromosomal ParB structure is replaced by an additional β-strand in the CTDs of P1 ParB and SopB. Analytical ultracentrifugation, native mass spectrometry and circular dichroism (CD) thermal melt scans further confirmed that the CTD was primarily dimeric in solution and measured a $T_m$ of 68°C (*Figure 3—figure supplement 1E and F*). NMR H-D exchange data revealed that the dimer exchanges slowly (which will be relevant to the interpretation of later experiments), with those most stable being the intermolecular H-bonds between the two β2 strands (data not shown; the half-lives range from 10 min to 4 hr). This secondary structure element is at the centre of the hydrophobic core which is

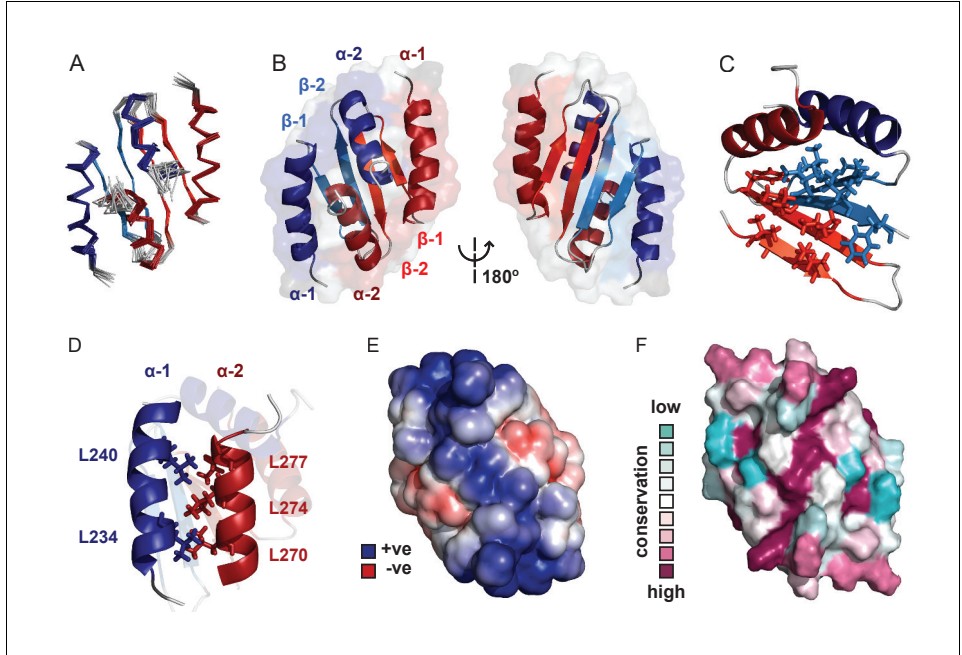

**Figure 3.** Solution NMR structure of the dimeric ParB C-terminal domain. (A) Ensemble overlay of the 14 lowest-energy CTD structures. Red and blue depict separate monomers within the dimer. (B) Secondary structure elements are identified. α1 indicates the N-terminus of each monomer. (C) The hydrophobic core of Ile, Val, Leu and Phe residues. For clarity, portions of both monomers were removed. (D) Interdigitating Leu residues of both monomers form a leucine-zipper interaction. (E) Surface charge representation reveals a large electropositive region across the β-sheet face (orientation as in B, right hand side). Continuum electrostatics calculations used the PDB2PQR web server (*Dolinsky et al., 2004*) and the APBS plugin for PyMOL (*Lerner and Carlson, 2006*; *Baker et al., 2001*). (F) Evolutionary conservation surface profile of the CTD of ParB prepared using ConSurf (*Goldenberg et al., 2009*; *Celniker et al., 2013*) (orientation as in E). The chemical shifts, restraints and structural co-ordinates have been deposited with the BMRB (34122) and PDB (5NOC).

DOI: https://doi.org/10.7554/eLife.28086.006

The following source data and figure supplement are available for figure 3:

**Source data 1.** NMR assignment, structure calculation and validation statistics.
DOI: https://doi.org/10.7554/eLife.28086.008

**Figure supplement 1.** Solution NMR structure of the dimeric ParB CTD domain.
DOI: https://doi.org/10.7554/eLife.28086.007

made up of several Ile, Val and Phe residues in the β-sheet (*Figure 3C*). The α1 helix forms a leucine zipper with the α2 helix, where alternating Leu residues interdigitate (*Figure 3D*). A striking feature of the structure is a highly electropositive face of the dimer arising from several conserved Lys residues (*Figure 3E and F*) analogous to the plasmid-encoded SopB and P1 ParB proteins (*Figure 3— figure supplement 1G,H and I*).

## The CTD binds DNA non-specifically via a lysine-rich surface

To test the idea that the lysine-rich surface we had observed might bind to DNA, we performed TBE-EMSAs with the isolated CTD. These showed that the CTD was indeed able to bind dsDNA (*Figure 4Ai*) resulting in the formation of a 'ladder' of bands of decreasing mobility. This is highly reminiscent of patterns formed by full length ParB under the same conditions (*Figure 4Aii* and (*Taylor et al., 2015*)) except for the presence of smaller gaps between the 'rungs' as would be expected for a protein of smaller size. The CTD was also shown to bind to hairpin oligonucleotides as short as 10 bp and to ssDNA (*Figure 4—figure supplement 1A* and data not shown). We do not see substantial differences in the affinity of ParB for DNA substrates with different sequences and so this binding activity appears to be non-specific (data not shown). Native mass spectrometry of complexes formed between the CTD and a 15 bp duplex DNA revealed a stoichiometry of 1 DNA per dimer (*Figure 4D*). This is in contrast to the P1 ParB system where the CTD operates in a different binding mode, and can bind two 16-mers (*Schumacher et al., 2007*).

To further probe the putative DNA binding surface, we performed a titration of the 10 bp hairpin DNA against the isotopically-labelled CTD dimer (*Figure 4E*, assigned $^1$H-$^{15}$N HSQC spectra are shown in *Figure 4—figure supplement 1B*). Residues with large chemical shift perturbations (CSPs, $\Delta\delta > 0.08$) are either directly involved in DNA-binding or undergo a conformational change as an indirect result of DNA-binding, and these were mapped onto the structure (*Figure 4F*). Two regions of interest were identified: D231-V233, and K252-K259, which are found on the intermolecular β-sheet face and proximal loop regions to form a large, concave and positively-charged interaction surface (*Figure 4G*).

To confirm that this surface was responsible for DNA binding we substituted several Lys residues with Ala and monitored the effect on DNA binding using EMSA and PIFE assays. In the first instance, a dual K255A/K257A substitution was studied in the context of both the CTD-only construct (CTD$^{KK}$) and the full length ParB protein (ParB$^{KK}$). CTD$^{KK}$ displayed a greatly reduced affinity (~50 fold) for DNA, but the binding was not completely abolished (*Figure 4Bi*). CD thermal melt analysis confirmed that this defect was not attributable to global misfolding (*Figure 4—figure supplement 1C*). The full length ParB$^{KK}$ variant showed a sigmoidal DNA binding isotherm in a PIFE assay, indicating strong positive cooperativity as observed for wild type ParB but the apparent $K_d$ was 6-fold weaker (*Figure 4—figure supplement 1D*). In EMSA assays, this variant showed no defect in specific binding to *parS* as would be expected (*Figure 4—figure supplement 1E*). Somewhat more surprisingly however, these lysine substitutions appeared to have a negligible effect on nsDNA binding when assessed using the TBE-EMSA assays (*Figure 4Bii*). This may well reflect the complexity that arises when a partially defective nsDNA binding locus is physically attached to the wild type CDBD domain (which is still fully competent to bind DNA).

We next designed a triple K252A/K255A/K259A variant with the aim of fully dissipating the positive charge density across the surface of the CTD, rather than only targeting the loop-proximal regions. EMSA analysis showed that DNA binding was completely abolished in CTD$^{KKK}$ up to concentrations of 50 μM (*Figure 4Ci* and *Figure 4—figure supplement 1F*). CD thermal melt analysis showed that CTD$^{KKK}$ was equivalently folded to wild type CTD at ambient temperatures, but with a reduced $T_m$ (53°C) indicating a moderate destabilising effect of the mutations (*Figure 4—figure supplement 1G*). Interestingly, analysis of full length ParB$^{KKK}$ showed a clear and consistent defect in all nsDNA binding assays used. TBM-EMSA gels showed that ParB-*parS* complexes were still formed, although it should be noted that their yield was reduced relative to wild type (*Figure 4—figure supplement 1I*). TBE-EMSA gels showed a complete eradication of the discrete lower mobility bands which arise from nsDNA binding (*Figure 4Cii*). Moreover, nsDNA binding was undetectable using the PIFE analysis (*Figure 4—figure supplement 1H*). Interestingly, EMSA analysis showed that DNA-bound ParB$^{KKK}$ networks do still form as very low mobility species that assemble co-operatively at high ParB concentrations. Given that the ParB$^{KKK}$ protein retains a functional HtH motif, this property might well reflect a role for the CDBD in binding to nsDNA, albeit weakly compared to the *parS*

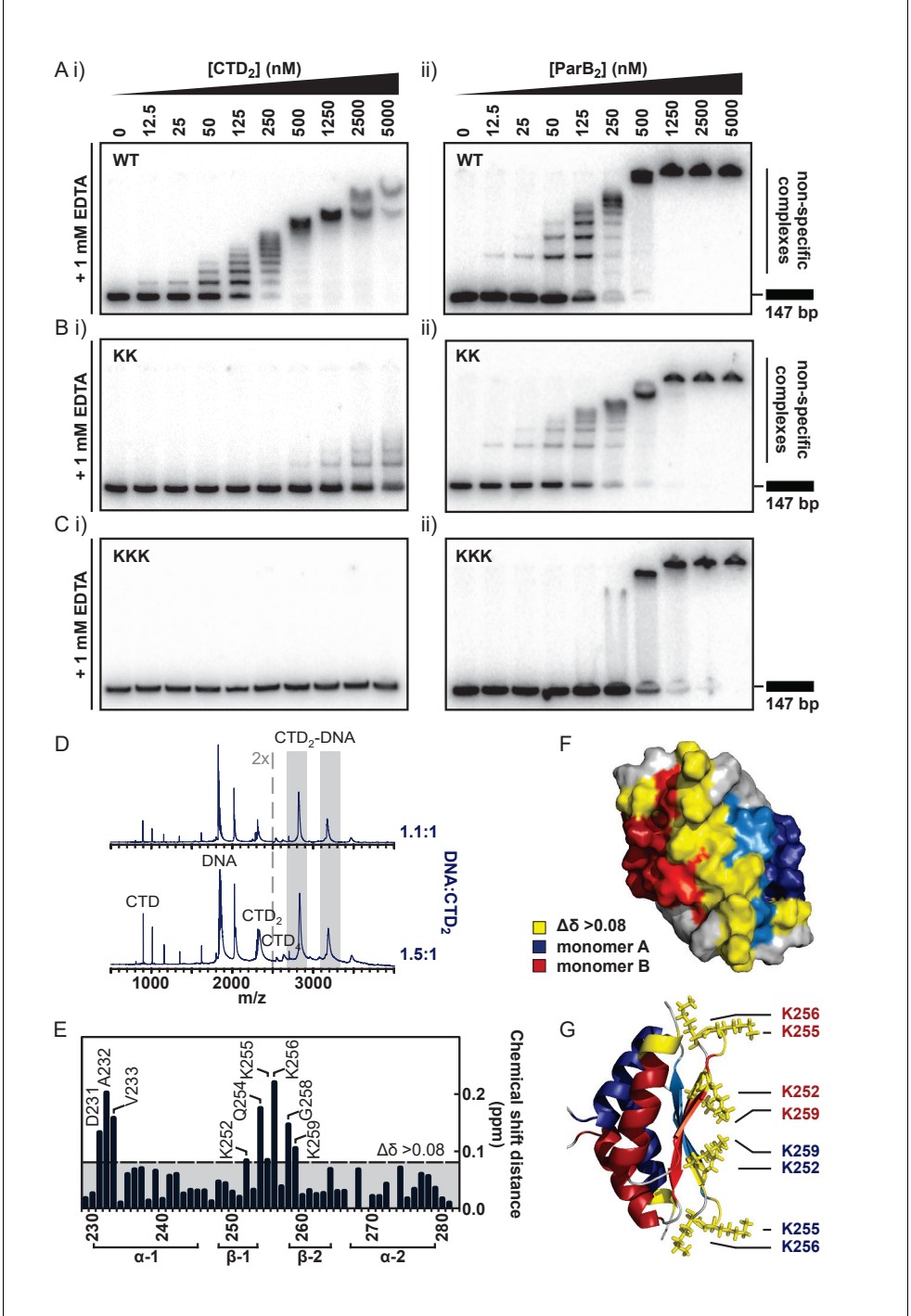

**Figure 4.** The CTD binds DNA via a lysine-rich surface. (**A–C**) TBE-EMSAs for the titration of full length ParB and CTD against 147 bp DNA. Wild type and mutant proteins, K255A + K257A and K252A + K255A + K259A, are indicated. (**D**) Native mass spectrometry. Titrations of a 15 bp nsDNA hairpin against CTD were performed between ratios of 1.1:1 and 1.5:1 DNA:CTD$_2$. Example spectra are shown with the DNA-CTD$_2$ complex shaded in grey. (**E**) Deviations in the assigned $^1$H-$^{15}$N HSQC spectra of CTD upon titration with a 10 bp hairpin DNA. (**F**) Chemical shift perturbations exceeding 0.08 Δδ are highlighted on the structure in yellow. (**G**) Lys residues thought most likely to bind to DNA are shown as sticks.

DOI: https://doi.org/10.7554/eLife.28086.009

The following figure supplement is available for figure 4:

**Figure supplement 1.** The CTD binds DNA via a lysine-rich surface.

*Figure 4 continued on next page*

*Figure 4 continued*

DOI: https://doi.org/10.7554/eLife.28086.010

sequence. This highlights the potential complexity of nsDNA binding in ParB which might involve synergistic binding by both the CDBD and CTD domains.

## DNA binding by the CTD is essential for DNA condensation and bridging in vitro

We next exploited our double- and triple-lysine mutant ParB proteins to test the role of the DNA-binding activity associated with the CTD in forming condensed ParB networks. These networks have been extensively characterised previously for wild type ParB using magnetic tweezers with single tethered DNA substrates (*Taylor et al., 2015*) and also in TIRF-based microscopy (*Graham et al., 2014*).

Unlike full length ParB, the CTD was not capable of condensing DNA tethers under any condition tested, even up to 5 µM $CTD_2$ concentrations and under applied forces as low as 0.02 pN (*Figure 1—figure supplement 1Ai*). This is consistent with the expected requirement for multiple protein-protein and/or protein-DNA interfaces to promote DNA looping and condensation. Incubation of full length $ParB^{KK}$ with single DNA tethers resulted in defective DNA condensation compared to wild type ParB (*Figure 5A*). When it was observed, condensation was sudden (rather than progressive, as for wild type ParB) and full condensation required the applied force to be dropped to an exceptionally low value (0.09 pN) (*Figure 5—figure supplement 1Aii*). The DNA molecules also showed unusually large steps when decondensed by force, suggesting that $ParB^{KK}$ was infrequently stabilising *in cis* DNA-bridging interactions between isolated DNA regions (data not shown). Co-incubation of $ParB^{KKK}$ with single DNA tethers under our standard experimental conditions resulted in no measurable condensation events, even under applied forces as low as 0.02 pN and at elevated concentrations (*Figure 5A* and *Figure 5—figure supplement 1Aiii*). The average work done by ParB compared to the variant proteins during these condensation events was determined from the difference between the integral of the force-extension curve in the presence of the protein and that of DNA alone. This provides a means to quantitatively compare the condensation efficiency between mutants (*Figure 5—figure supplement 1B*). We also performed plectoneme stabilisation experiments (*Figure 5B*). In this assay, a single torsionally-constrained DNA molecule is positively supercoiled at a 4 pN force by applying 60 turns. ParB is then introduced and, after full buffer exchange, all turns are released whilst monitoring DNA extension. Any deviation of DNA extension from that expected of bare DNA is indicative of supercoiled regions being stabilised by ParB. $ParB^{KK}$ could stabilise DNA-bridging interactions between isolated DNA regions but this was often characterised by large steps in the DNA tether extension increase which is unlike the behaviour of wild type (*Figure 5C* and *Figure 5—figure supplement 1C*). $ParB^{KKK}$ was unable to stabilise plectoneme structures showing that it cannot bridge DNA segments *in trans* (*Figure 5D* and *Figure 5—figure supplement 1D*).

## The CTD can both inhibit the formation of, and decondense, ParB-DNA networks in vitro

The CTD potentially acts as both an oligomerisation interface and also a site of nsDNA-binding. Therefore, we hypothesised that the CTD might have a dominant negative effect on full length ParB by competing for the DNA and protein interfaces that mediate the formation of ParB networks in the magnetic tweezers (MT).

Purified CTD completely inhibited the formation of the condensed state if pre-incubated with wild type ParB and DNA under the high stretching force regime (*Figure 6A* and *Figure 6—figure supplement 1A*). We also tested whether the introduction of free CTD to pre-condensed tethers was able to disrupt ParB-DNA networks. Condensed ParB networks were completely stable in a flow of free ParB on the timescale of these experiments, and the DNA tethers were also able to re-condense following force-induced decondensation (*Figure 6Bi*). However, the inclusion of excess free CTD rapidly disrupted ParB networks, with some degree of decondensation observed in 94% of all the molecules tested (*Figure 6Bii*). Moreover, those molecules which did not decondense

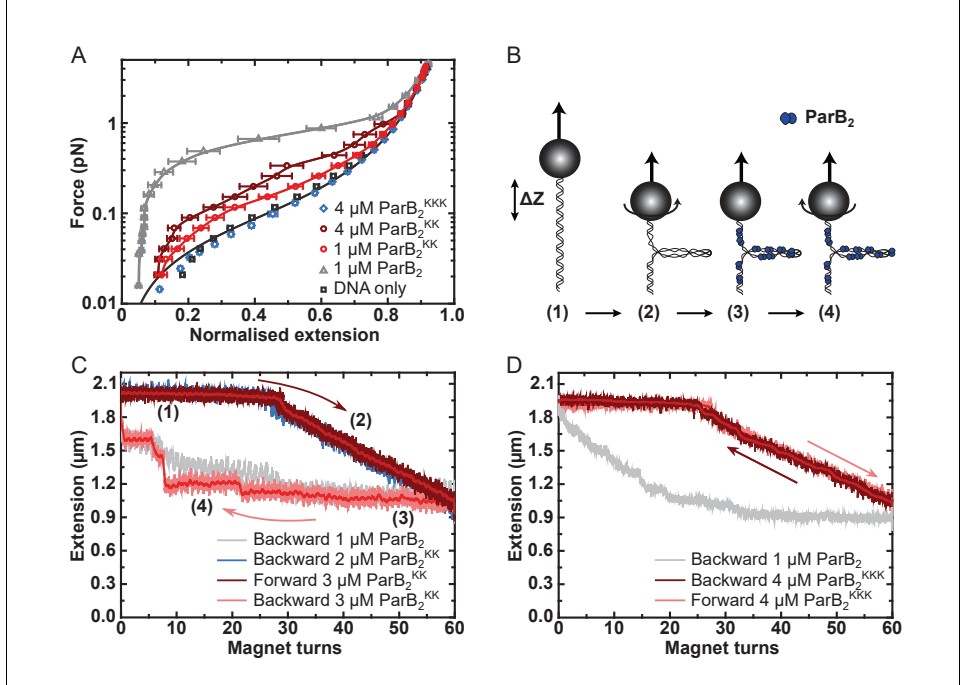

**Figure 5.** DNA binding by the CTD is required for efficient DNA condensation in vitro. (**A**) Mean force-extension curves of DNA molecules co-incubated with ParB variants at the indicated concentrations. Non-condensed (protein-free) DNA data is fitted to the worm-like chain model. Solid lines in condensed data are guides for the eye. Errors are the standard error of the mean of measurements on different molecules (N ~ 18–35 molecules). (**B**) Schematic of plectoneme stabilisation assay. A single torsionally-constrained DNA molecule was positively supercoiled at 4 pN force by applying 60 turns. This shortens the tether length due to the formation of plectonemes in the overwound DNA. $ParB_2$ is then introduced and all turns are released whilst monitoring DNA extension. Evidence for ParB-dependent plectoneme stabilisation is provided by hysteresis in the extension as a function of magnet turns as the supercoiling is removed. (**C**) Plectoneme stabilisation assay comparing bare DNA, wild type ParB and $ParB^{KK}$. The double-mutant protein supported DNA bridging and occasionally large steps were observed in the backward trace (see text for discussion). (**D**) Plectoneme stabilisation assay comparing wild type ParB and $ParB^{KKK}$. No activity was detected for the triple-mutant protein.

DOI: https://doi.org/10.7554/eLife.28086.011

The following figure supplement is available for figure 5:

**Figure supplement 1.** DNA binding by the CTD is required for efficient DNA condensation in vitro.

DOI: https://doi.org/10.7554/eLife.28086.012

spontaneously could be stretched by force, but were then unable to recondense when permissive forces were restored. This ability of the CTD to decondense ParB networks demonstrates that the protein-protein and/or protein:DNA interfaces that maintain the condensed state under a low force regime are dynamic (i.e. they are exchanging while the overall structure of the network is maintained).

We have shown above that the CTD binds tightly to nsDNA. Therefore, its ability to prevent condensation and induce decondensation might simply reflect competition for the nsDNA that becomes available during exchange of ParB:DNA interfaces. Indeed, we have shown previously that free DNA is a potent inducer of network decondensation in the MT apparatus (*Taylor et al., 2015*). To test the idea that the CTD dimerisation interface is also important for maintaining the condensed state, we repeated our experiments with the $CTD^{KK}$ and $CTD^{KKK}$ constructs, which are defective and apparently unable (respectively) to bind nsDNA. Both mutant proteins were as effective as wild type in preventing condensation (*Figure 6C* and *Figure 6—figure supplement 1B*), and both were able to induce decondensation in approximately 95% of all molecules tested (*Figure 6Biii and D*). This strongly suggests that CTD-dependent ParB network dissipation is primarily mediated by competition for the CTD dimerisation interface and further confirms that the $CTD^{KKK}$ construct is folded. This

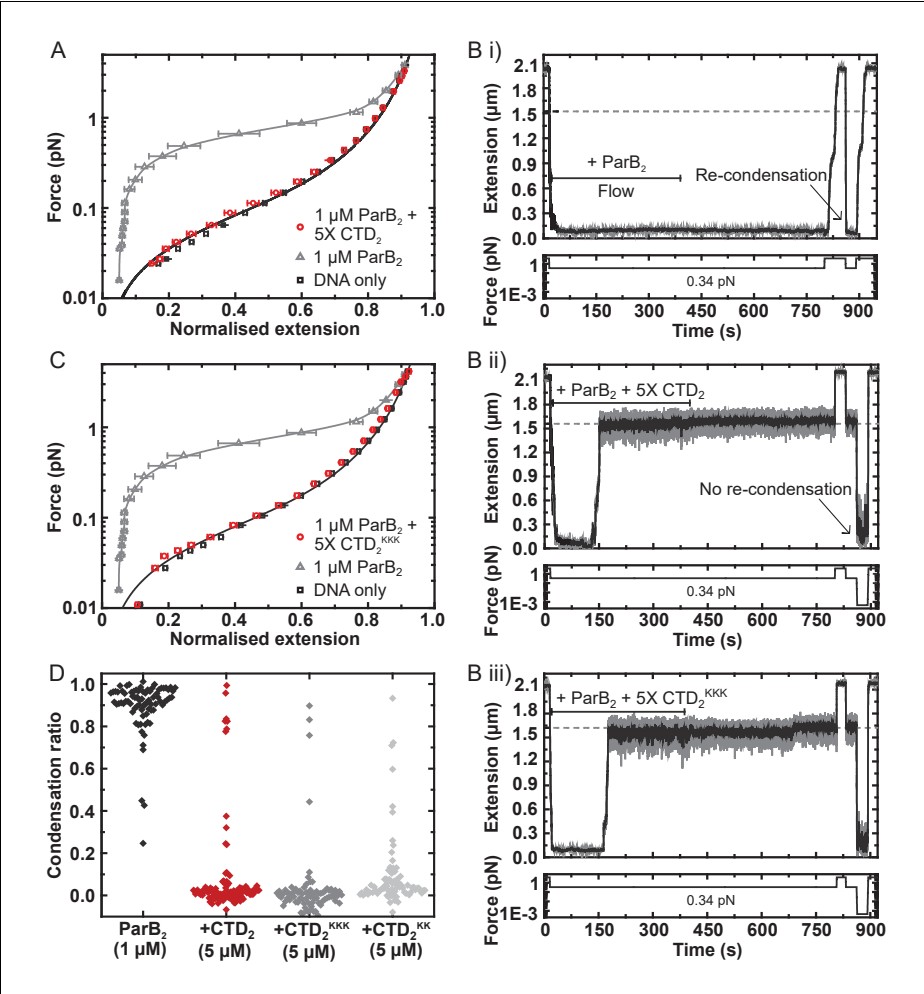

**Figure 6.** The CTD of ParB both inhibits and disrupts ParB-dependent DNA condensation. (**A**) Mean force-extension curves for DNA molecules co-incubated with 1 μM ParB$_2$ in the presence or absence of 5 μM CTD$_2$. Non-condensed (protein-free) DNA data is fitted to the worm-like chain model. Solid lines in condensed data are guides for the eye. (**B**) (i) ParB-DNA networks are stable in magnetic tweezers in the presence of 1 μM ParB$_2$. (ii) ParB-DNA complexes spontaneously decondense following the introduction of 1 μM ParB$_2$ and 5 μM free CTD$_2$. (iii) ParB-DNA complexes spontaneously decondense following the introduction of 1 μM ParB$_2$ and 5 μM CTD$_2$$^{KKK}$. (**C**) Mean force-extension curves for DNA molecules co-incubated with 1 μM ParB$_2$ in the presence or absence of 5 μM mutant CTD$_2$$^{KKK}$. Errors are the mean of measurements on different molecules (N > 40 molecules) (**D**) Condensation ratio (see Materials and methods for definition) for individual DNA condensation events involving the addition of CTD competitor variants to pre-condensed ParB-DNA networks.
DOI: https://doi.org/10.7554/eLife.28086.013

The following figure supplements are available for figure 6:

**Figure supplement 1.** The CTD of ParB both inhibits and disrupts ParB-dependent DNA condensation.
DOI: https://doi.org/10.7554/eLife.28086.014

**Figure supplement 2.** Possible mechanisms to explain the dominant negative effect of the CTD on full length ParB.
DOI: https://doi.org/10.7554/eLife.28086.015

---

competition presumably results from the formation of heteroligomers between full length ParB and the CTD, which disrupts interactions that are essential for condensation (*Figure 6—figure supplement 1C*).

## The CTD is critical for the formation of ParB foci in vivo

To test the importance of the CTD dimerisation and DNA binding interfaces in vivo, we compared the ability of wild type and mutant ParB-GFP proteins to form foci in *B. subtilis* cells when expressed from the endogenous locus. Wild type ParB-GFP formed discrete foci around *oriC* as expected (*Murray et al., 2006*; *Graham et al., 2014*; *Autret et al., 2001*; *Real et al., 2005*; *Glaser et al., 1997*; *Lin et al., 1997*; *Lewis and Errington, 1997*; *Marston and Errington, 1999*). In contrast, ParB$^{KKK}$-GFP failed to form discrete foci (*Figure 7Ai–ii*) despite wild type expression (*Figure 7—figure supplement 1A*). Interestingly, the triple-mutant protein appeared to localise non-specifically to the nucleoid, perhaps as a result of residual DNA binding by the HtH motifs, suggesting that ParB$^{KKK}$-GFP retained the ability to dimerise. A caveat in interpreting this experiment is that, in addition to a complete eradication of nsDNA binding by the CTD domain, the ParB$^{KKK}$ mutant protein also showed a reduction in *parS* binding (*Figure 4—figure supplement 1I*). Therefore, we cannot exclude the possibility that defective *parS* binding also contributes to the lack of ParB foci formation we have observed. A ParB$^{L270D+L274D}$ construct, designed to prevent leucine zipper-mediated dimerisation of the CTD, was completely unable to form ParB foci (*Figure 7Aiii–v*) despite being expressed at approximately wild type levels (*Figure 7—figure supplement 1B*). The complete deletion of the CTD by truncation to E222 or E227 resulted in the same phenotype (data not shown).

Our attempts to purify recombinant ParB$^{L270D+L274D}$ failed because the protein was insoluble upon overexpression in *E. coli*. This raises the possible caveat that the loss of function associated with this dimerisation mutant in vivo might reflect mis-folding. Therefore, we also investigated whether the free CTD was able to interfere with dimerisation in vivo, thereby causing a dominant negative effect on ParB function. A *B. subtilis* strain was engineered with a C-terminal *gfp* fusion replacing the endogenous *spo0J* and the unlabelled CTD-only gene inserted at an ectopic locus downstream of a P$_{hyperspank}$ promoter, designed for high protein expression that is tightly-controlled with IPTG (*Figure 7B*). Overexpression of the CTD caused ParB-GFP foci to become diffuse (*Figure 7C*), although expression levels of endogenous ParB-GFP were unaffected (*Figure 7—figure supplement 1C*).

ChIP-qPCR analysis allowed us to more directly characterise the effect of CTD expression upon ParB spreading (i.e. the enrichment of ParB at and widely around *parS* sites). Spreading was measured around a single *parS* site (359.20°) and used a locus towards the terminus to monitor background 'enrichment' (146.52°) (*Figure 7D*). As expected, in the absence of CTD expression, ParB was highly enriched not only at *parS* sites (~40 fold), but also for several kilobase pairs around *parS* (*Figure 7E*). Overexpression of CTD significantly decreased the signal around *parS* (up to ~4 fold), indicating that it interferes with spreading. Western blotting of cells grown under equivalent conditions to the ChIP-qPCR assay and using the same batch of polyclonal anti-ParB antibody suggests that the CTD is not preferentially recognised over the endogenous full length ParB protein (*Figure 7—figure supplement 1D*). Note that the reduced signal observed for the *parS* fragment does not necessarily indicate defective specific binding, because the PCR product at *parS* is much larger than the 16 bp *parS* site or the 24 bp footprint of a ParB dimer (*Murray et al., 2006*). We can conclude that non-specific DNA interactions are reduced, but we are unable to say whether specific interactions are also reduced, or are maintained at wild type levels.

Finally, we determined the consequence of decreased ParB spreading in vivo induced by CTD overexpression by measuring the rate of DNA replication initiation. ParB normally inhibits the activity of Soj, a regulator of the master bacterial initiation protein DnaA (*Murray and Errington, 2008*). Marker frequency analysis showed that CTD overexpression stimulated the frequency of DNA replication initiation, indicating that regulation of DnaA by Soj was adversely affected (*Figure 7—figure supplement 1E*). Together, these results are consistent with our in vitro observations, and support a model in which dynamic ParB-DNA networks are dependent upon ParB oligomerisation and DNA-binding interfaces in the CTD.

## Discussion

ParB proteins form long-distance bridging interactions on DNA, forming foci that facilitate chromosomal partitioning reactions (*Lynch and Wang, 1995*; *Rodionov et al., 1999*; *Bingle et al., 2005*). These foci are anchored at *parS* sites and interact non-specifically around a single site for ±18 kbp (*Breier and Grossman, 2007*; *Wang et al., 2017*). This ParB 'spreading' activity is a conserved

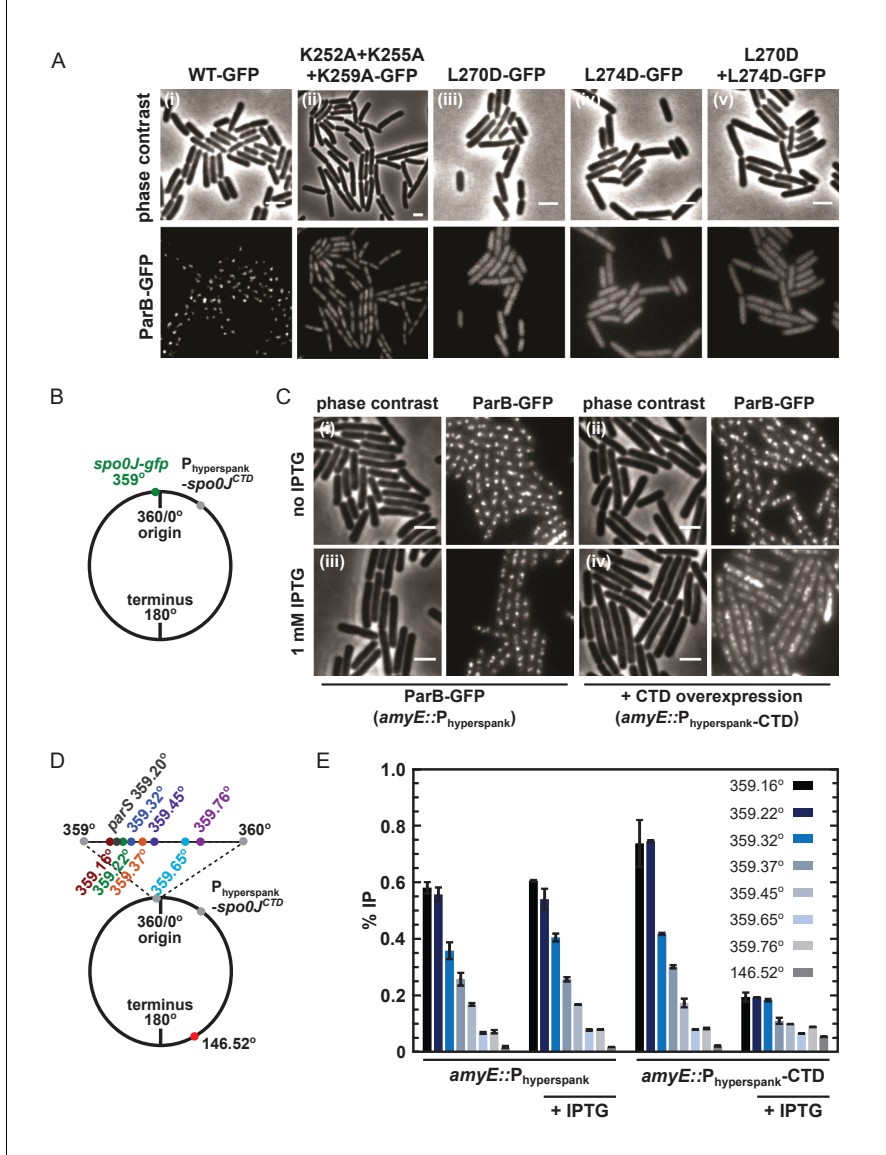

**Figure 7.** DNA-binding and dimerisation by the CTD is critical for ParB function in vivo. (**A**) Variant ParB-GFP mutants form abnormal foci in *B. subtilis*. Cells were grown overnight in slow growth conditions before dilution (1:100) into fast growth media, and were allowed to achieve at least five mass doublings before observation by microscopy (N > 300 cells). Scale bar is 2 μm. (**B**) Construct design for overexpression of the CTD in vivo. (**C**) CTD overexpression was induced by IPTG in the presence of chromosomally-encoded wild type ParB-GFP using the $P_{hyperspank}$ construct. Cells were grown as in A. (**D**) Construct design for ChIP-qPCR. (**E**) ChIP-qPCR assay for ParB spreading. Cells were grown slowly overnight, diluted (1:100) into fast growth media, and allowed to reach eight mass doublings before crosslinking with 1% formaldehyde. Background IP was measured at the terminus (146°). Primer pairs produced 200–300 bp fragments.

DOI: https://doi.org/10.7554/eLife.28086.016

The following figure supplement is available for figure 7:

**Figure supplement 1.** DNA-binding and dimerisation by the CTD is critical for ParB function in vivo.
DOI: https://doi.org/10.7554/eLife.28086.017

property across chromosomal and plasmid segrosomes, yet the interaction interfaces involved have remained elusive, particularly for genomically-encoded systems (*Graham et al., 2014*). This is, in part, due to the variable structures of ParB proteins and their cognate centromere sequences, even within the type I subclass of which *B. subtilis* ParB is a member (*Gerdes et al., 2000*;

*Schumacher, 2008*). Increasing evidence indicates that ParB spreading is the result of a DNA-bridging activity mediated by ParB-ParB oligomerisation interfaces (*Graham et al., 2014*; *Taylor et al., 2015*; *Broedersz et al., 2014*; *Sanchez et al., 2015*; *Sanchez et al., 2013*). However, a complete understanding of the relationship between ParB structure and function has been hindered by the lack of any full length structure for a chromosomally-encoded ParB. Indeed, the organisation of the N-terminal (NTD), central DNA-binding (CDBD) and C-terminal (CTD) domains appears to be quite complex (*Figure 1—figure supplement 1A,B and C*; *Figure 6—figure supplement 2A*). For the type I ParB protein class, there is evidence to suggest that dimerisation and/or tetramerisation can occur at the NTD and CDBD, and that dimerisation can occur at the CTD (*Leonard et al., 2004*; *Schumacher and Funnell, 2005*; *Chen et al., 2015*). Alongside an ability to bind DNA both specifically and non-specifically, a combination of some or all of these protein:protein interfaces must support ParB spreading and network formation (*Figure 6—figure supplement 2A*).

To address the putative role of the CTD in spreading (*Leonard et al., 2004*), we resolved the first structure of a genomic ParB CTD. The structure revealed a conserved lysine-rich surface and we showed that this binds to DNA in an apparently non-specific manner. This novel DNA binding locus is distinct from the sequence-specific DNA binding site for *parS* formed by the classical HtH motif within the CDBD domain. In this respect, there are parallels with the plasmid-encoded ParB proteins P1 and SopB (*Schumacher et al., 2007*; *Schumacher et al., 2010*). In these systems, the CTDs share similar surface electrostatics in which a polar distribution of charged residues results in both positively- and negatively- charged surfaces on opposite faces of the domain. In both *B. subtilis* ParB and P1 ParB, the Lys/Arg rich surface has been shown to bind to DNA using structural or biochemical techniques ([*Schumacher and Funnell, 2005*; *Schumacher et al., 2007*] and this work), but experiments with SopB do not support the idea that it shares this activity (*Ah-Seng et al., 2009*). The integrity of the CTD may also be important for stabilising the N-terminal region of the protein. When SopB was truncated ahead of the CTD, it could not bind *sopC* (the F-plasmid equivalent of *parS*) (*Kusukawa et al., 1987*), and analogous results have been obtained with *T. thermophilus* and *B. subtilis* ParB ((*Leonard et al., 2004*) and unpublished observations).

Our CTD structure facilitated the design of separation of function mutations to test the importance of the dimerisation and DNA binding activities using a variety of in vitro and in vivo readouts of ParB function. We showed that the DNA binding interface in the CTD is not required for *parS* binding, and that this is instead dependent on the HtH motif found within the CDBD domain as predicted in several previous studies (*Graham et al., 2014*; *Chen et al., 2015*; *Autret et al., 2001*; *Gruber and Errington, 2009*; *Gerdes et al., 2000*; *Bignell and Thomas, 2001*; *Theophilus and Thomas, 1987*; *Lobocka and Yarmolinsky, 1996*). In contrast, the CTD is essential for the formation of nsDNA complexes that are observed as ladders of decreasing mobility in EMSA assays. Mutant proteins that were unable to bind DNA at the CTD locus were severely defective in both DNA condensation assays in vitro and ParB foci formation assays in vivo. Moreover, ParB proteins that were designed to be unable to form oligomeric structures by mutation of the CTD-CTD dimerisation interface were completely unable to form ParB foci. Finally, we showed that the free CTD domain can disrupt ParB networks, both in vitro and in vivo. Overexpression of the CTD in *B. subtilis* can lead directly to the assembly of heterodimers or heterooligomers with full length ParB that interfere with wild-type function. However, this is not the case when using purified proteins in vitro, and we envision at least two non-exclusive explanations for the CTD-induced decondensation we observe in the MT assay (see *Figure 6—figure supplement 2B*): the CTD could simply compete for binding to the DNA substrate (scenario 1 in the figure), or the CTD could exchange with full length ParB to form mixed species (scenarios 2 and 3). The first possibility can be excluded as being solely responsible for decondensation, because a CTD construct that is devoid of DNA binding activity has the same dominant negative effect. In the second scenario, the CTD might form inactive heterodimers with the full length ParB in free solution, such that it can longer exchange with the condensed network. Alternatively, the CTD could interact with the full length ParB that remains bound to DNA in the network. This could 'cap' the bridging interactions, if indeed the CTD interface were important for such interactions (although we cannot exclude the alternative idea that binding of the CTD has an allosteric effect on a different bridging interface). In either case our results highlight a critical function for the CTD interface in overall ParB function, but we favour the second idea. Although our NMR experiments indicate that exchange of the CTD interface does occur, it is on a timescale of hours to days. This is too slow to account for the effect we observe in the MT if it is simply based on monomer

exchange between the CTD dimer and the full length ParB dimer *in free solution*. However, the situation in the ParB network is very different, because the opposing force generated by the MT can dramatically increase the rate of exchange of bridging interactions that hold the DNA in a condensed state. Ultimately, a direct demonstration of this idea will require a correlative measurement of DNA extension with the observation of ParB/CTD binding under conditions of controlled force, and this will be the subject of future studies. Taken together, our observations support the idea that the CTD is essential for both DNA binding and for ParB-ParB bridging interactions that support DNA condensation in vitro.

We propose that the presence of two DNA binding loci in ParB can help to explain how ParB networks are anchored at *parS* in vivo. Importantly, this architecture resolves the paradoxical observation that the apparent specificity for *parS* in vitro (<10 fold greater affinity for *parS* versus nsDNA) is insufficient to explain the strict localisation of ParB around just eight sites in a ~4 Mbp genome (*Taylor et al., 2015*; *Broedersz et al., 2014*). In a two DNA binding site model, specific and non-specific binding can be semi-independent activities that are architecturally-coupled only when ParB oligomerises into networks. This model can also explain why DNA condensation does not require *parS* in vitro, whereas the absence of *parS* sites prevents the formation of ParB-DNA foci in vivo ((*Graham et al., 2014*; *Taylor et al., 2015*; *Sanchez et al., 2015*; *Erdmann et al., 1999*) and this work). In a test tube, whenever ParB is present at concentrations that licence oligomerisation, it is always in large stoichiometric excess over binding sites and all available DNA will be bound. In cells, the situation is very different because there is a limited pool of ParB (*Graham et al., 2014*). Specific interaction with *parS* preferentially anchors the ParB network at *parS,* leaving a vast number of unoccupied sites. If *parS* sites are absent in cells, ParB might still form networks, but these would not be anchored at specific sites and would therefore fail to form foci, as has been observed experimentally (*Erdmann et al., 1999*; *Sullivan et al., 2009*). A rigorous proof of these ideas will require a modelling approach that will be the subject of future work.

Previously, high-resolution SIM and ChIP-seq data have suggested that ParB-DNA partition complexes involve stochastic and dynamic binding of ParB to both DNA and other ParB proteins, resulting in the formation of fluid intra-nucleoid 'ParB cages' on DNA (*Sanchez et al., 2015*). This view is consistent with the disorder observed in MT assays (*Taylor et al., 2015*), and with the dominant negative effect of the free CTD domain on ParB networks shown here. However, a recent structural study of *H. pylori* ParB concluded that a novel tetramerisation interface within the NTD was also likely to be important in bridging (*Chen et al., 2015*; *Song et al., 2017*). Moreover, spreading could be facilitated by *parS*-dependent conformational changes that act as nucleation points for networks (*Broedersz et al., 2014*; *Leonard et al., 2004*). A more complete understanding of ParB network formation and its regulation will be required to underpin future studies on how ParB acts together with ParA and condensin to orchestrate efficient chromosome segregation.

## Materials and methods

### Plasmids and DNA substrate preparation

All mutagenesis used the pET28a-ParB expression vector as a template (*Taylor et al., 2015*). The R149G mutation was introduced by site-directed mutagenesis using a QuikChange II XL kit (Agilent Technologies). The full length ParB gene (1-849) with the K255A + K257A or K252A + K255A + K259A substitutions was produced synthetically (Life Technologies) and subcloned into pET28a using NcoI and BamHI restriction sites (*Taylor et al., 2015*). CTD only (217-282) constructs were produced using PCR with primer overhangs incorporating 5' PacI and 3' XmaI restriction sites for subcloning (5' - GCGTAAGCCCCGGGCAGAATGTTCCACGTGAAACAAAG - 3' and 5' - GCGTCATGTTAATTAATCATTATGATTCTCGTTCAGACAAAAG - 3') into pET47b (Novagen) to produce a protein with an N-terminal HRV 3C protease cleavable His-tag. The integrity of all DNA sequences was confirmed by direct sequencing (DNA Sequencing Service, University of Dundee).

Preparation of radiolabelled, 5' Cy3-labelled and magnetic tweezer DNA substrates was as described (*Taylor et al., 2015*). 10 bp DNA hairpins were prepared by heating a self-complementary oligonucleotide (5' - GCGTACATCATTCCCTGATGTACGC - 3') in 10 mM $Na_2HPO_4$/$NaH_2PO_4$ pH 6.5, 250 mM KCl and 5 mM EDTA to 95°C for 25 min, followed by rapid cooling in an ice bath. The DNA was purified by anion-exchange chromatography using a 0.25–1 M KCl gradient, and desalted

over multiple NAP-10 columns (GE Healthcare Life Sciences) before concentration in a centrifugal vacuum concentrator.

## ParB overexpression and purification

ParB, and the variants R149G, K255A + K257A and K252A + K255A + K259A, were overexpressed and purified as described (*Taylor et al., 2015*). CTD, and mutants thereof, were His-tagged and purified to homogeneity as follows. Cell pellets, produced as described (*Taylor et al., 2015*), were resuspended in 20 mM Tris-HCl pH 7.5, 500 mM NaCl and 1 mM BME (TNB buffer) with the addition of 10 mM imidazole, 5% (v/v) glycerol and protease inhibitor cocktail set II (Millipore), before being snap-frozen and stored at −80°C. Cells were lysed by sonication in the presence of 0.2 mg/ml lysozyme (Sigma). The lysate was clarified by centrifugation and loaded onto a 5 ml HisTrap HP column (GE Healthcare Life Sciences) equilibrated with TNB buffer +10 mM imidazole. CTD elution was achieved with a linear gradient of 10 mM to 500 mM imidazole. Peak fractions were assessed by SDS-PAGE and pooled accordingly. The tag was removed with HRV 3C protease (Thermo Scientific, Pierce) for 16 hr at 4°C during dialysis into TNB buffer +10 mM imidazole. The products were subsequently loaded onto a HisTrap HP column whereby the cleaved CTD was collected in the flow-through volume, followed by concentration by centrifugation in Amicon Ultra-15 3 kDa MWCO spin filters (Millipore). This concentrate was loaded at 1 ml/min onto a Hiload 16/600 Superdex S75 gel filtration column (GE Healthcare Life Sciences) equilibrated with 50 mM Tris-HCl pH 7.5, 1 mM EDTA, 300 mM NaCl and 1 mM DTT (storage buffer). Appropriate peak fractions were pooled, followed by final concentration by centrifugation as described. Spectrophotometric grade glycerol (Alfa Aesar) was added to 10% (v/v). The final protein was then snap-frozen as aliquots and stored at −80°C. ParB concentration was determined by spectrophotometry using theoretical extinction coefficients of 7450 $M^{-1}$ $cm^{-1}$ and 2560 $M^{-1}$ $cm^{-1}$ for ParB and CTD respectively. ParB concentrations in all assays refer to the dimeric state. The wild type, K255A + K257A and K252A + K255A + K259A variants of ParB behaved equivalently during purification, and run almost identically on preparative size exclusion columns (both in the context of the CTD and full length protein), suggesting that they are all dimeric. For structure determination by NMR, the CTD was dual isotopically ($^{13}$C and $^{15}$N) labelled during overexpression in M9 media, as described previously (*Williams et al., 2007*), and subsequently purified as above.

## CD spectroscopy

CD spectra were collected using a JASCO J-810 spectropolarimeter fitted with a Peltier temperature control (Jasco UK). 50 μM protein samples were buffer exchanged into phosphate buffered saline (PBS; 8.2 mM $NaH_2PO_4$, 1.8 mM $KH_2PO_4$, 137 mM NaCl and 2.7 mM KCl (pH 7.4)) by 16 hr dialysis at 4°C using a membrane with a MWCO of 3.5 kDa. At 20 μM and using a 0.1 cm quartz cuvette, thermal stability data was acquired across a 190–260 nm absorbance scan (1 nm data pitch at a scanning speed of 100 nm/min) from 5°C to 90°C at 5°C increments. Raw data was normalised to molar ellipticity (MRE ($deg.cm^2.dmol^{-1}$)) using calculation of the concentration of peptide bonds and the cell path length. A buffer only baseline was subtracted from all datasets. All data for mutant variants was acquired alongside a wild type CTD control.

## NMR

NMR datasets were collected at 35°C, utilising a Varian VNMRS 600 MHz spectrometer with a cryogenic cold-probe. The purified protein was buffer exchanged into PBS (10 mM $NaH_2PO_4$, 1.8 mM $KH_2PO_4$, 137 mM NaCl and 2.7 mM KCl (pH 6.1)) and concentrated to 1 mM. $^1$H-$^{15}$N HSQC, $^1$H-$^{13}$C HSQC, HNCACB, CBCA(CO)NH, HNCA, HN(CO)CA, HNCO, CC(CO)NH, H(CCO)NH, HCCH-TOCSY, $^{15}$N-NOESY-HSQC (150 ms mixing time), $^{13}$C-NOESY-HSQC (140 ms mixing time) and aromatic $^{13}$C-NOESY-HSQC (140 ms mixing time) experiments were collected on $^{13}$C,$^{15}$N-labelled CTD. 2D $^1$H-$^1$H TOCSY and NOESY spectra were recorded on unlabelled protein. $^{13}$C,$^{15}$N-labelled and unlabelled protein were mixed in equimolar amounts to create a mixed labelled sample used to record 3D $^{13}$C,$^{15}$N F$_1$-filtered, $^{13}$C,$^{15}$N F$_3$-edited $^{13}$C-NOSEY-HSQC and $^{15}$N-NOESY-HSQC experiments (*Zwahlen et al., 1997*). A hydrogen-deuterium (HD) exchange experiment was conducted by recording $^1$H-$^{15}$N HSQC experiments at several intervals following dissolution of freeze-dried protein in $D_2O$. A titration was conducted by adding a 10 bp DNA hairpin step-wise to $^{13}$C, $^{15}$N-labelled

CTD and recording a $^1$H-$^{15}$N HSQC experiment after each addition. The final molar ratio of protein: DNA was 1:1.25. All NMR data were processed using NMRPipe (*Delaglio et al., 1995*). Spectra were assigned using CcpNmr Analysis 2.4 (*Vranken et al., 2005*). Proton chemical shifts were referenced with respect to the water signal relative to DSS.

Heteronuclear NOE experiments showed residues 214–228 to be highly flexible. This was supported by chemical shift analysis with TALOS+ (*Shen et al., 2009*) and the absence of any medium or long-range NOEs for these residues. Structure calculations were only conducted on residues 229–280, as the unstructured tail made unfavourable energy contributions to the calculation which distorted the selection of ensembles of low-energy structures. Structure calculations were conducted using ARIA 2.3 (*Rieping et al., 2007*). 10 structures were calculated at each iteration except iteration 8, at which 200 structures were calculated. The 20 lowest energy structures from this iteration went on to be water refined. Spin diffusion correction was used during all iterations (*Linge et al., 2004*). Two cooling phases, each with 30,000 steps were used. Torsion angle restraints were calculated using TALOS+. Standard ARIA symmetry restraints for two monomers with C2 symmetry were included (*Bardiaux et al., 2009*). Structural rules were enabled, using the secondary structure predictions made by TALOS+. The HD exchange experiment showed 29 NH groups to be protected after 8 min. Initial structure calculations were conducted without hydrogen bond restraints. Hydrogen bond donors were then identified and corresponding hydrogen bond restraints included in later calculations. Calculations were conducted using a flat-bottom harmonic wall energy potential for the distance restraints until no consistent violations above 0.1 Å were observed. The final calculation was then performed using a log-harmonic potential (*Nilges et al., 2008*) with a softened force-field (*Mareuil et al., 2015*). Structures were validated using the Protein Structure Validation Software (PSVS) suite 1.5 (*Bhattacharya et al., 2007*) and CING (*Doreleijers et al., 2012*). The chemical shifts, restraints and structural co-ordinates have been deposited with the BMRB (34122) and PDB (5NOC).

## EMSA experiments

The specific and nsDNA-binding activity of ParB was analysed by TBM- and TBE-PAGE as described (*Taylor et al., 2015*). Serial dilutions of ParB, to the indicated concentrations, were incubated with 20 nM 147 bp *parS* or 'scrambled' DNA (at a ratio of 1:19 labelled to unlabelled), 50 mM HEPES-KOH pH 7.5, 100 mM KCl, 2.5 mM MgCl$_2$, 0.1 mg/ml BSA, 1 mM DTT and 2.5% (v/v) Ficoll in a 20 µl reaction volume. Where indicated, different length dsDNA substrates were used equivalently. Samples were incubated at room temperature for 30 min followed by 5 min on ice. 10 µl of each were loaded onto a 6% acrylamide/bis-acrylamide (29:1) gel in 90 mM Tris, 150 mM H$_3$BO$_3$ (final pH 7.5), supplemented with either 2.5 mM MgCl$_2$ (TBM) or 1 mM EDTA (TBE). Gels were pre-run at 150 V, 4°C for 30 min in a buffer identical to their composition, and run post-loading at 150 V, 4°C for 1 hr. For imaging, gels were vacuum-dried, exposed to a phosphor screen and subsequently scanned by a Phosphor-Imager (Typhoon FLA 9500, GE Healthcare Life Sciences). Gels are representative of three biological replicate experiments.

## Protein induced fluorescence enhancement (PIFE) assay

ParB DNA-binding to non-specific substrates was analysed in a solution-based assay where a change of emitted Cy3 fluorescence acted as a reporter of ParB binding (*Taylor et al., 2015*). ParB was incubated with 20 nM 147 bp 5'-Cy3-labelled DNA, 50 mM HEPES-KOH pH 7.5, 100 mM KCl, 2.5 mM MgCl$_2$, 0.1 mg/ml BSA and 1 mM DTT. Samples of 120 µl were incubated at room temperature for 30 min before being transferred into a quartz cuvette for data collection. Cy3 fluorescence in each sample was measured by excitation at 549 nm and an emission scan between 560 and 600 nm (Cary Eclipse Fluorescence Spectrophotometer, Agilent Technologies). Peak maxima were calculated by the area under the curve function in GraphPad Prism software, and the increase in fluorescence calculated relative to a DNA only control. Where appropriate, data was fitted with a Hill equation.

$$Y = \frac{B_{max} \times [ParB_2]^h}{Kd_{app}^h + [ParB_2]^h} \tag{1}$$

Where Y is the measured increase in fluorescence, B$_{max}$ is the maximal increase in fluorescence, h is the Hill coefficient and Kd$_{app}$ is the apparent dissociation constant. When comparing wild type and mutant binding isotherms, the data were well-fitted using a shared value for the Hill coefficient

(i.e. there was no evidence for changes in binding cooperativity as a result of the mutations studied). Standard errors for the fitted parameters were calculated in GraphPad Prism using three independent repeat experiments.

## Magnetic tweezers

### Instrument and samples

We used a home-made magnetic tweezers setup similar in design to that described in *Strick et al. (1998)*, *Carrasco et al. (2013)* and detailed in *Pastrana et al. (2016)*. In brief, images of 1 μm superparamagnetic beads tethered to the surface of a glass slide by DNA constructs are acquired with a 100x oil immersion objective and a CCD camera. Real-time image analysis was used to determine the spatial coordinates of beads with nm accuracy in *x*, *y* and *z*. A step-by-step motor located above the sample moves a pair of magnets allowing the application of stretching forces to the bead-DNA system. Applied forces can be quantified from the Brownian excursions of the bead and the extension of the DNA tether. Unless specified otherwise, data were acquired at 120 Hz and filtered down to 3 Hz for representation and analysis.

Fabrication of DNA substrates for MT experiments containing a single *parS* sequence with biotins and digoxigenins at the tails was described in *Taylor et al. (2015)*. The DNA substrates were incubated with 1 μm streptavidin-coated beads (MyOne, Invitrogen) for 10 min. Then, the DNA-bead complex was injected in a liquid cell functionalised with anti-digoxigenin antibodies (Roche) and incubated for 10 min before applying force. In a first step, visual inspection allows identification and selection of tethered DNA molecules. Torsionally-constrained molecules and beads with more than a single DNA molecule were identified from its distinct rotation-extension curves. Double or multiple tethers were discarded for further analysis in this work. All the experiments were performed in a reaction buffer composed of 100 mM NaCl, 50 mM Tris-HCl or HEPES-KOH pH 7.5, 100 μg/ml BSA and 0.1% Tween 20 (v/v).

### CTD-induced decondensation

Once selected single torsionally-relaxed DNA molecules, 1 μM $ParB_2$ was incubated for 2–3 min and condensation was induced by decreasing the force to 0.34 pN. Immediately after this, one cell-volume of reaction buffer containing 5 μM CTD and 1 μM $ParB_2$, pre-incubated for 2–3 min, was applied at a constant flow velocity of 16 μl/min. In control experiments where only 1 μM $ParB_2$ was applied, the reaction was supplemented with a volume of storage buffer equal to that used in the CTD experiments and thus maintaining the ionic conditions.

To have a measurement of the degree of induced decondensation, we determined a condensation ratio, $C_r$ (*Figure 6D*), which was calculated simply as:

$$C_r = \frac{z_0 - z}{z_0} \tag{2}$$

where $z_0$ is the expected extension at 0.34 pN measured before ParB injection and $z$ is the equilibrium extension after induced-decondensation. $z$ was determined from average extensions of 120 data points at 390 s after the cell volume was completely exchanged. These data were acquired at 60 Hz and filtered down to 3 Hz.

### Force-extension curves and work calculation

Force-extension curves were obtained by decreasing the applied force in steps from 4 pN to ~0.02 pN for a total measuring time of 13 min. This procedure is initially performed for bare DNA molecules. Then, the force is reset to 4 pN and ParB variants are flown and incubated for 2 min before starting the measurement of a new force-extension curve using the same magnet positions in absence of proteins. In every case, the force applied to each bead was calculated from the force-extension data of bare DNA molecules.

The work done during condensation ($\Delta W$) can be calculated by the difference in work between the force-extension curve in the presence of ParB variants and that of bare DNA (*Equation 3*), where $z_{max}$ is the extension at the maximum applied force of 4 pN. Integrals were calculated using the trapezoidal rule using OriginLab software.

$$\Delta W = \int_0^{z_{max}} (F_{ParB} - F_{DNA})dz \qquad (3)$$

## Native mass spectrometry

Mass spectra were collected using a Synapt G2 HDMS T-wave ion mobility mass spectrometer (Waters) with nano-electrospray using *in-house* made gold-coated borosilicate capillaries. Protein only experiments required buffer exchange to 300 mM $NH_4Ac$ (pH 6.9) using Micro Bio-Spin P-6 Gel columns (Bio-Rad). In analysis of CTD-DNA interactions, mixtures of CTD and 15 bp hairpin DNA were co-incubated for 5 min at 30°C prior to buffer exchange. The sequence of the 15 bp DNA hairpin was 5'- GCATAGCGTACATCATTCCCTGATGTACGCTATGC-3'. CTD samples were loaded at 10 or 50 µM. The following parameters were applied to preserve non-covalent interactions (*Sobott et al., 2005*; *Konijnenberg et al., 2013*): backing pressure ca. 5 mbar (adjusted with Speedivalve), source pressure $5.8 \times 10^{-3}$ mbar, trap pressure $4.4 \times 10^{-2}$ mbar; capillary voltage 1.3–1.7 kV, sampling cone 20–60 V, extraction cone 1 V, trap and transfer collision energy 10–25 V and 2–5 V, trap DC bias 35–45 V, IMS wave velocity 300–750 m/s, IMS wave height 40 V, helium cell gas flow 180 ml/min, IMS gas flow 90 ml/min (IMS gas cell pressure ca. 3.1 mbar) and source temperature 30°C. The measured mass of CTD was 8096.1 ± 0.2 Da, which matches well with the calculated value of 8096.1 Da. Molecular weights of multiply charged proteins, DNA and complexes were calculated using the MaxEnt1 function in MassLynx (Waters). For the error of the mass measurements in both directions, the MaxEnt peak width at half height was divided by 2. Both biological (new sample preparations from a fresh stock aliquot) and technical (repeat MS measurements of the same buffer exchanged complexes) repeats were undertaken.

## AUC

Sedimentation velocity experiments were conducted in an Optima XL-A analytical ultracentrifuge using an An-60 Ti rotor (Beckman) at 20°C. 420 µl volume solutions of 250 µM ParB CTD were prepared in storage buffer with 10% (v/v) glycerol and loaded into a sedimentation velocity cell with sapphire windows and a buffer only reference channel. A rotor speed of 60,000 rpm was employed, with absorbance scans ($A_{280}$) taken across a radial range of 5.85 to 7.25 cm at 2 min intervals to a total of 200 scans. Data were fitted (baseline, meniscus, frictional coefficient (f0), and time- and radial-invariant noise) to a continuous c(s) distribution model using SEDFIT version 9.4, at a 95% confidence level (*Schuck, 2000*; *Brown and Schuck, 2006*). The partial specific volume ($\bar{v}$) of the CTD and storage buffer density and viscosity were calculated using Sednterp (*Hayes et al., 1995*). Residuals are shown as a grayscale bitmap where the vertical axis lists each of the 200 scans (with scan one at the top) and the horizontal axis depicts radial position over which the data were fitted. Shade indicates variance between fitted and raw data.

## In vivo fluorescence imaging

Nutrient agar (Oxoid) was used for routine selection and maintenance of *B. subtilis* strains. Supplements were added as required: chloramphenicol (5 µg/ml), erythromycin (1 µg/ml), kanamycin (2 µg/ml), spectinomycin (50 µg/ml), tetracycline (10 µg/ml), zeomycin (10 µg/ml), and ampicillin (200 µg/ml). Cells were grown in defined minimal medium base (Spizizen minimal salts supplemented with Fe-$NH_4$-citrate (1 µg/ml), $MgSO_4$ (6 mM), $CaCl_2$ (100 µM), $MnSO_4$ (130 µM), $ZnCl_2$ (1 µM), thiamine (2 µM)) supplemented with casein hydrolysate (200 µg/ml) and/or various carbon sources (succinate (2.0%), glucose (2.0%)). Supplements were added as required: tryptophan (20 µg/ml), erythromycin (1 µg/ml), spectinomycin (50 µg/ml), IPTG (1 mM). Standard techniques were used for strain construction (*Harwood and Cutting, 1990*). *B. subtilis* competent cells were transformed using an optimised two-step starvation procedure as described (*Anagnostopoulos and Spizizen, 1961*; *Hamoen et al., 2002*). All plasmids and strains were verified by sequencing.

To visualise cells, starter cultures were grown at 37°C overnight in SMM-based medium supplemented with tryptophan (20 µg/ml), casein hydrolysate (200 µg/ml), succinate (2.0%), then diluted 1:100 into fresh medium supplemented with glucose (2.0%) and with/without 1 mM IPTG (as indicated) and allowed to achieve early exponential growth ($OD_{600}$ 0.3–0.4). Cells were mounted on ~1.2% agar pads (0.25X minimal medium base) and a glass coverslip was placed on top. To visualise individual cells the cell membrane was stained with 0.4 µg/ml FM5-95 (Molecular Probes).

Microscopy was performed on an inverted epifluorescence microscope (Nikon Ti) fitted with a Plan-Apochromat objective (Nikon DM 100x/1.40 Oil Ph3). Light was transmitted from a 300 Watt xenon arc-lamp through a liquid light guide (Sutter Instruments) and images were collected using a CoolSnap HQ$^2$ cooled CCD camera (Photometrics). All filters were Modified Magnetron ET Sets from Chroma. Digital images were acquired of >300 cells per sample (and for two biological repeats) and analysed using METAMORPH software (version V.6.2r6).

## ChIP-qPCR

To determine the amount of ParB bound to the chromosome by ChIP-qPCR, starter cultures were grown overnight at 30°C in SMM-based medium supplemented with tryptophan (20 μg/ml), casein hydrolysate (200 μg/ml) and succinate (2.0%), then diluted 1:100 into fresh medium supplemented with glucose (2.0%) and 1 mM IPTG (as indicated), and allowed to grow to an $A_{600}$ of 1. Samples were treated with sodium phosphate (final concentration 10 mM) and cross-linked with formaldehyde (final concentration 1%) for 10 min at room temperature, followed by a further incubation for 30 mins at 4°C. Cells were pelleted at 15°C and washed three times with PBS (pH 7.3). Cell pellets were resuspended in 500 μl of lysis buffer (50 mM NaCl, 10 mM Tris-HCl pH 8.0, 20% sucrose, 10 mM EDTA, 100 μg/ml RNase A, ¼ complete mini protease inhibitor tablet (Roche), 2000 K u/μl Ready-Lyse lysozyme (Epicentre)) and incubated at 37°C for 45 min to degrade the cell wall. 500 μl of IP buffer (300 mM NaCl, 100 mM Tris-HCl pH 7.0, 2% Triton X-100, ¼ complete mini protease inhibitor tablet (Roche), 1 mM EDTA) was added to lyse the cells and the mixture was incubated at 37°C for a further 10 min before cooling on ice for 5 min. To shear DNA to an average size of ~500 to 1000 bp samples were sonicated (40 amp) four times at 4°C. The cell debris was removed by centrifugation at 15°C and the supernatant transferred to a fresh Eppendorf tube. To determine the relative amount of DNA immunoprecipitated compared to the total amount of DNA, 100 μl of supernatant was removed, treated with Pronase (0.5 mg/ml) for 10 min at 37°C before SDS (final concentration 0.67%) was added, and stored at 4°C.

To immunoprecipate protein-DNA complexes, 800 μl of the remaining supernatant was incubated with polyclonal anti-ParB antibodies (Eurogentec) for 1 hr at room temperature. Protein-G Dynabeads (750 μg, Invitrogen) were equilibrated by washing with bead buffer (100 mM Na$_3$PO$_4$, 0.01% Tween 20), resuspended in 50 μl of bead buffer, and then incubated with the sample supernatant for 3 hr at room temperature. The immunoprecipated complexes were collected by applying the mixture to a magnet and washed once with the following buffers for 5 min in the respective order: 0.5X IP buffer; 0.5X IP buffer + NaCl (500 mM); stringent wash buffer (250 mM LiCl, 10 mM Tris-HCl pH 8.0, 0.5% Tergitol-type NP-40, 0.5% C$_{24}$H$_{39}$NaO$_4$, 10 mM EDTA). Finally, the complexes were washed a further three times with TET buffer (10 mM Tris-HCl pH 8.0, 1 mM EDTA, 0.01% Tween 20) and resuspended in 112 μl of TE buffer (10 mM Tris-HCl pH 8.0, 1 mM EDTA). Formaldehyde crosslinks of both the total DNA and the immunoprecipate was reversed by incubation at 65°C for 16 hr. The reversed DNA was then removed from the magnetic beads and transferred to a clean PCR tube and stored at 4°C for qPCR analysis.

To measure the amount of DNA bound to ParB, GoTaq (Promega) qPCR mix was used for the PCR reactions and qPCR was performed in a Rotor-Gene Q Instrument (Qiagen) using serial dilution of the immunoprecipitate and the total DNA control as the template. Oligonucleotide primers were then designed that amplify at an interval of ~500–1000 bp away from $parS^{359°}$ and were typically 20–25 bases in length and amplified a ~ 200–300 bp PCR product (*Table 1*). Error bars indicate the standard deviation of two technical replicates.

## Western blot analysis

Proteins of the whole cell extract were separated by electrophoresis using a NuPAGE 4–12% Bis-Tris gradient gel in MES buffer (Life Technologies) and transferred to a Hybond-P PVDF membrane (GE Healthcare Life Sciences) using a semi dry apparatus (Hoefer Scientific Instruments). Polyclonal primary antibodies were used to probe protein of interest and then detected with an anti-rabbit horseradish peroxidase-linked secondary antibody using an ImageQuant LAS 4000 mini digital imaging system (GE Healthcare Life Sciences).

**Table 1.** Primer sequences used in ChIP-qPCR

| Plasmid | Genotype | Genome location |
| --- | --- | --- |
| oqAKPCR3 | 5'-AGCCGGATTGATCAAACATC-3' | 359.32° |
| oqAKPCR4 | 5'-AGAGCCGATCAGACGAAAAC-3' | 359.32° |
| oqAKPCR5 | 5'-GAGGCAAGCAAAGCTCACTC-3' | 359.45° |
| oqAKPCR6 | 5'-TGCCATGACAGAGCTGAAAC-3' | 359.45° |
| oqAKPCR7 | 5'-CTTTTCCAAGGCCTTTAGCC-3' | 359.22° |
| oqAKPCR8 | 5'-TCACGGAAAAACCCATCATTT-3' | 359.22° |
| oqAKPCR9 | 5'-TATTGGCCTGCTTCATACCC-3' | 359.65° |
| oqAKPCR10 | 5'-TGGAGATTCTGTCCACGAAA-3' | 359.65° |
| oqPCR9 | 5'-AAAAAGTGATTGCGGAGCAG-3' | 359.16° |
| oqPCR10 | 5'-AGAACCGCATCTTTCACAGG-3' | 359.16° |
| oqPCR25 | 5'-TCCATAATCGCCTCTTGGAC-3' | 359.37° |
| oqPCR26 | 5'-AAGCGCATGCTTATGCTAGG-3' | 359.37° |
| oqPCR31 | 5'-GATCCGAAGGTCTGTCTACG-3' | 359.76° |
| oqPCR32 | 5'-CGATTGCGATTGTACGGTTG-3' | 359.76° |
| oqPCR57 | 5'-TTTGCATGAACTGGGCAATA-3' | 146.52° |
| oqPCR58 | 5'-TCCGAACATGTCCAATGAGA-3' | 146.52° |

DOI: https://doi.org/10.7554/eLife.28086.018

## Marker frequency analysis

To obtain chromosomal DNA, starter cultures were grown at 37°C in SMM based medium supplemented with tryptophan (20 µg/ml), casein hydrolysate (200 µg/ml), succinate (2.0%) overnight, then diluted 1:100 into fresh medium supplemented with glucose (2.0%) and with/without 1 mM IPTG (as indicated) and allowed to achieve early exponential growth ($OD_{600}$ 0.3–0.5). Sodium azide (0.5%; Sigma) was added to exponentially growing cells to prevent further metabolism. Chromosomal DNA was isolated using a DNeasy Blood and Tissue Kit (Qiagen). GoTaq (Promega) qPCR mix was used for PCR reactions. Q-PCR was performed in a Rotor-Gene Q Instrument (Qiagen). For quantification of the origin, the intergenic region between *dnaA* and *dnaN* was amplified using primers 5'-GATCAATCGGGGAAAGTGTG-3' and 5'-GTAGGGCCTGTGGATTTGTG-3'. For quantification of the terminus, the region downstream of *yocG* was amplified using primers 5'-TCCATATCCTCGCTCCTACG-3' and 5'-ATTCTGCTGATGTGCAATGG-3'. By use of crossing points ($C_T$) and PCR efficiency a relative quantification analysis ($\Delta\Delta C_T$) was performed using Rotor-Gene Software version 2.0.2 (Qiagen) to determine the *ori/ter* ratio of each sample. These results were normalised to the *ori/ter* ratio of a DNA sample from *B. subtilis* spores which only contain one chromosome and thus have an *ori/ter* ratio of 1. Error bars indicate the standard deviation of three technical replicates.

## Acknowledgements

We thank Benjamin Bardiaux for providing the Aria software, and also Chris Williams, Joe Beesley and Antony Burton for assistance with NMR, CD and AUC data acquisition, respectively. GLMF was supported by the Biotechnology and Biological Sciences Research Council (1363883). MSD, JAT, VAH and TDC were supported by the Wellcome Trust (100401 and 077368). FMH acknowledges support from the European Research Council (681299) and from MINECO (FIS2014-58328-P). HM was supported by The Royal Society.

## Additional information

### Funding

| Funder | Grant reference number | Author |
|---|---|---|
| Wellcome | 100401 and 077368 | Victoria A Higman<br>James A Taylor<br>Timothy Craggs<br>Mark Simon Dillingham |
| Biotechnology and Biological Sciences Research Council | 1363883 | Gemma LM Fisher |
| H2020 European Research Council | 681299 | Fernando Moreno-Herrero |
| Ministerio de Economía y Competitividad | FIS2014-58328-P | Fernando Moreno-Herrero |
| Royal Society | | Heath Murray |

The funders had no role in study design, data collection and interpretation, or the decision to submit the work for publication.

### Author contributions

Gemma LM Fisher, Conceptualization, Data curation, Formal analysis, Investigation, Methodology, Writing—original draft, Writing—review and editing; César L Pastrana, Alan Koh, Annika Butterer, Data curation, Formal analysis, Investigation, Methodology, Writing—review and editing; Victoria A Higman, Data curation, Formal analysis, Validation, Investigation, Methodology, Writing—review and editing; James A Taylor, Data curation, Formal analysis, Investigation, Writing—review and editing; Timothy Craggs, Formal analysis, Supervision, Methodology, Writing—review and editing; Frank Sobott, Resources, Supervision, Funding acquisition, Methodology, Writing—review and editing; Heath Murray, Resources, Supervision, Funding acquisition, Methodology, Writing—original draft, Writing—review and editing; Matthew P Crump, Resources, Supervision, Methodology, Writing—review and editing; Fernando Moreno-Herrero, Conceptualization, Supervision, Funding acquisition, Methodology, Writing—review and editing; Mark S Dillingham, Conceptualization, Supervision, Funding acquisition, Methodology, Writing—original draft, Project administration, Writing—review and editing

### Author ORCIDs

Gemma LM Fisher (iD) http://orcid.org/0000-0001-8468-5032
Timothy Craggs (iD) http://orcid.org/0000-0002-7121-0609
Mark S Dillingham (iD) http://orcid.org/0000-0002-4612-7141

### Decision letter and Author response

Decision letter https://doi.org/10.7554/eLife.28086.024
Author response https://doi.org/10.7554/eLife.28086.025

## Additional files

### Supplementary files

• Transparent reporting form
DOI: https://doi.org/10.7554/eLife.28086.019

### Major datasets

The following dataset was generated:

| Author(s) | Year | Dataset title | Dataset URL | Database, license, and accessibility information |
|---|---|---|---|---|
| Victoria A Higman, Gemma LM Fisher, Mark Simon Dillingham, Matthew P Crump | 2017 | Solution NMR Structure of the C-terminal domain of ParB (Spo0J) | https://www.rcsb.org/pdb/explore/explore.do?structureId=5NOC | Publicly available at the RCSB Protein Data Bank (accession no: 5NOC) |

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
