## [Decision Letter]

Thank you for submitting your article "The structural basis for dynamic DNA binding and bridging interactions which condense the bacterial centromere" for consideration by *eLife*. Your article has been favorably evaluated by John Kuriyan (Senior Editor) and three reviewers, one of whom is a member of our Board of Reviewing Editors. The reviewers have opted to remain anonymous.

The reviewers have discussed the reviews with one another and the Reviewing Editor has drafted this decision to help you prepare a revised submission.

Summary:

Using a combination of structural and single-molecule biophysical approaches, the authors study the structure of the *B. subtilis* ParB protein (Spo0J), how it interacts with specific and nonspecific DNA, and how it forms supramolecular structures. In general, the experiments are well designed and multiple approaches were used to examine DNA binding and condensation. The structure of the *B. subtilis* Spo0J C-terminal domain (CTD) is new. Particularly, the authors demonstrate that the central DNA-binding domain (CDBD) is required for specific DNA binding but not needed for interaction with nonspecific DNA and that the CTD is responsible for the intramolecular dimerization and the binding to nonspecific DNA. They specifically distinguish between the CTD-DNA and CTD-CTD interactions by using a CTD mutant that does not bind DNA but is shown to interfere with compaction. The mechanism proposed explains a large amount of previously reported data and represents a significant step forward in our understanding of bacterial chromosome segregation. However, before publication is considered, additional experimental data are needed to address the concerns detailed below.

Essential revisions:

1) The CTD-CTD interaction:

The authors use the CTD peptide both in in vitro competition and in vivo interference assays. They observe an inhibition of DNA condensation in vitro, and the absence of foci formation in vivo, from which they conclude on the direct involvement of the CTD in these activities. However, another interpretation of the results would be that the CTD from the peptide and the full-length exchange between dimers resulting in equilibrium with heterodimers as the major species. In particular, in vivo this would lead to an important decrease in ParB2 concentration that would greatly reduce the possibility to spread on the DNA and thus to form visible foci. Moreover, the overexpression of CTD reduces significantly the specific binding of ParB to *parS*, while the spreading activity in the vicinity of *parS* is not reduced (Figure 5). This data appears in direct contradiction with the author's proposition of the roles of the CTD for which we would expect no reduction in *parS* binding and the absence of spreading away from *parS*. In the same vein, in vitro, the formation of heterodimers would also prevent the formation of the DNA binding domain (alignment of the two HTH motifs in the ParB dimer) resulting in the absence of DNA compaction.

A major concern surrounds the implication that the C-terminal dimerization domain exists as a monomer (e.g. Figure 1) and/or can exchange during DNA binding (exchange is mentioned several times in the Results but not in the Discussion). As far as I know, in all ParBs tested at physiological, solution concentrations (low μm and nM), C-terminal fragments are always dimers and N-terminal fragments are never dimers. The N-terminal interactions have been uncovered only at crystallographic concentrations. In the third paragraph of the Introduction, for example, the compact dimer of *T. thermophilus* Spo0J referred to here is not seen in biochemical experiments in the same study – the N-terminal fragments are monomers in solution.

In contrast to the situation proposed, the P1 ParB C-terminal dimer interaction is very strong and requires denaturation with GuHCl and renaturation to allow exchange and heterodimerization (JBC 2000, 275:8213). Otherwise dimer exchange does not happen during the time scale of DNA binding (> many minutes), although may occur with longer incubations (and likely would if coexpressed in vivo). The NMR H-D experiment cited but not presented (subsection “The structure of *Bs*ParB CTD reveals a dimer with a putative DNA binding interface”) does not allow interpretation without knowing concentrations and time scales. If exchange is significantly slower than DNA binding, then the ability to exchange is not relevant to the model. Similarly, the competition experiments in Figure 6 seem to preincubate the CTD with ParB, so it is not clear for how much time exchange has been allowed to occur. I would expect exchange to occur eventually if the two proteins are mixed together in vitro or expressed together in vivo, but the question of time-scale is critical here. Finally, exchange is not necessary for the bridging interactions to be necessary for condensation, nor do the data exclude roles for the N-terminal region (which previous studies, e.g. Graham et al., 2014, also show). I think the authors need to support the idea of exchange (during DNA binding) by reporting kinetics. They could measure the kD of dimerization directly. They could show that with their ParB, heterodimer ParB/DNA complexes form readily by EMSA, for example. Or their model could be modified or described differently if I have misinterpreted the implications here.

2) The non-specific DNA binding activity:

- In the absence of a totally defective mutant in the HTH motif, the author could not discriminate between the contribution of the HTH and the CTD domains for non-specific DNA binding. They cannot formally exclude that the HTH is not providing the main non-specific DNA binding activity as observed in some plasmid counterparts. Also, with the ParB^R149G^ variant, they cannot conclude that "the HTH motif is dispensable for non-specific binding and condensation" as point mutants in the second helix of HTH usually remains fully proficient for non-specific DNA binding.

- The non-specific activity measured by EMSA corresponds to DNA that is stuck in the wells. This retention occurs at the same protein concentration for all tested variants (Figure 2). For the triple KKK mutant, which is proposed to be defective in ns-DNA binding activity, the same retention in the well is observed (Figure 4—figure supplement 1) and occurs at the same concentration as for the WT, thus preventing to conclude initially that this activity (retention in the wells) arises from the non-specific DNA binding activity of the CTD.

- in vivo, the triple KKK mutant does not form foci as the WT ParB does. However, this variant displays a specific DNA binding activity on *parS* DNA that is strongly reduced (> 20-fold; compare Figure 4 to Figure 2), preventing directly the nucleation at *parS* and subsequently the formation of the foci. This variant could thus not serve as validating in vivo that the loss of the positively charge residues in the CTD prevents forming the ParB foci.

- The authors use in their rational that ParB of plasmid P1 harbors a DNA binding activity in the CTD but this is due to an extra loop in this dimerization domain that binds specifically to direct repeats in the cognate *parS* sequence, and thus could not be compared per se to the putative non-specific DNA binding interface.

[Editors' note: further revisions were requested prior to acceptance, as described below.]

Thank you for resubmitting your work entitled "The structural basis for dynamic DNA binding and bridging interactions which condense the bacterial centromere" for further consideration at *eLife*. Your revised article has been favorably evaluated by John Kuriyan (Senior Editor), a Reviewing Editor, and two reviewers.

The manuscript has been improved but there are some remaining issues that need to be addressed before acceptance. The reviewers comment on the need for clarification and rephrasing of interpretations in a number of places of the manuscript. While these issues can likely be addressed without the need for new experiments, a revised manuscript will need to include the suggestions put forward by the reviewers.

*Reviewer #2:*

The manuscript has been improved by adding schematics for the binding interfaces and by expending the discussion on the networking interactions and on the free-CTD experiments (– Discussion, third paragraph). Also the authors have removed some over-interpretations as mentioned in their rebuttal. Lastly, some rewriting – for instance subsection “The CTD binds DNA non-specifically via a lysine-rich surface”, last paragraph; subsection “The CTD is critical for the formation of ParB foci in vivo”, first and third paragraphs – clarify the interpretation of the experiments.

Amongst the new experiments provided within the rebuttal, it seems to me that at least Figure 2Ai-ii and Figure 2Bii should be included in the manuscript as supplementary data to support the arguing on the low exchange rate between subunits (as fully discussed in the new version).

However, the authors do not provide a direct evidence for the respective contribution of the HTH and CTD motifs to non-specific DNA binding activity, while some rationales/arguments have not been changed thus some sentences are still misleading to me:

- "we hypothesised that ParB contains a second DNA binding locus for nsDNA that functions independently of the helix-turn-helix motif. This idea was attractive to us for several reasons". As acknowledged by the authors in their rebuttal, the binding mode is very different for ParB of P1 as it binds specifically to dedicated motifs of *parS*. But, the 3rd reason still suggests that the CTD of ParB P1 is a precedent for nsDNA binding despite this argument should be removed.

- "This is in contrast to the P1 ParB system where the CTD can bind two 16-mers". This comparison should not be made as this compares two different activities (specific versus non-specific binding) by two different interfaces (loops versus positively charged faces).

- "Together, these data show that mutation of the HtH motif effectively eliminates the ability of ParB to interact specifically with its cognate parS site, while nsDNA binding and condensation is relatively unaffected. This is consistent with the idea that nsDNA binding may occur at a second DNA binding locus." The specificity of binding is totally lost with the point mutation (R149G), but nothing is shown concerning the remaining non-specific DNA binding activity from the HtH motif. Thus, testing only for the loss of specific DNA binding could not provide a direct argument for searching a "second binding locus" for non-specific DNA binding and should not be written as such as a rational.

- "A striking feature of the structure is a highly electropositive face of the dimer arising from several conserved Lys residues analogous to the plasmid-encoded SopB and P1 ParB proteins". The authors have to discuss at some point the difference with SopB for which, despite the presence of the high electropositive face in the CTD, no residual non-specific DNA binding activity was reported in a triple HtH mutant by contrast to a point mutant in the HtH (equivalent to ParB-R149G) that eliminates the *parS* DNA binding but not nsDNA binding (Ah-seng et al., 2009), both mutants harboring the highly electropositive face of the CTD.

*Reviewer #3:*

In this revision, Fisher et al. have further clarified the details of the activities and structure of the ParB C-terminal dimerization domain (CTD) in DNA binding and condensation. With one exception (see #1), they have addressed my concerns here. In this study, the major new information is the identification of non-specific DNA binding activity of the CTD. In addition, they present the first structure of the CTD of a bacterial ParB, which turns out to be similar to those of two plasmid ParBs published previously. The authors use sophisticated biophysical approaches, such as magnetic tweezers, to examine the condensation (considered as bridging) activities of ParB and their dependence on the CTD, and these approaches are uncommon and an elegant way to dissect these complexes. The idea that the dimerization domain is involved in bridging and condensation is not new and has been proposed and observed in several studies (e.g. Graham et al., 2014; Sanchez et al., 2015; Schumacher and Funnell, 2005; Schumacher, Mansoor and Funnell, 2007), but these experiments directly probe the role of the CTD in condensation.

1) My major concern is with the models (represented in cartoons in Figure 1 and Figure 6—figure supplement 2, form Y), that the CTD is monomeric in solution, a concern from the original review, to which the authors did not respond here. In their figures, they represent ParB as dimerized either by CTD or NTD in solution (but not both). As mentioned in previous reviews, the dimer forms of Spo0J/ParB lacking the CTD were seen only in the crystal structures, and the *T. thermophilus* Spo0J fragment was shown in the same paper to be monomeric in solution. I also note that the DNA binding experiments in the latter study were also done at crystallographic, not physiological, concentrations of protein. As well as published experiments from other ParBs, the current authors' own experiments presented here show that the CTD exchanges extremely slowly in solution, evidence of a very tight kD. I accept their arguments here that exchange may be different when bound to DNA, but this is not represented in their cartoons. Similarly, the poor folding and insolubility of the CTD leucine mutants (dimer interface) also supports that the CTD is never "free".

This (CTD monomers in solution) may not be what the authors intend, but it is misleading and needs to be changed (or clarified) in these cartoons.

2) Are the CTD or ParB KK and KKK mutants still dimers? The NMR structure clearly predicts that these mutations do not damage the dimer interface, and I expect this to be the case, but this should be confirmed experimentally and reported. Otherwise the authors cannot conclude that the effects of these mutations can be directly attributed to DNA binding defects. Similarly, they argue that CTD interference is due to dimerization since the KKK mutants also interfere; this is a circular argument without independent evidence that the mutations do not interfere with dimerization.

3) "we showed that the CTD is not required for *parS* binding": where are the experiments in this study to support the conclusion; i.e. *parS* binding in the absence of the CTD? The ParB/CTD-KKK mutants presumably still dimerize (comment #2) and indeed they contribute to the affinity of *parS* binding. I think that they mean that the CTD DNA binding interface is not required for *parS* binding, which should be clarified.

4) Subsection “The R149 residue within the HTH motif is essential for specific binding to *parS*, but not required for non-specific binding and condensation”, last sentence: The lack of specific DNA binding by the R149G mutation does not mean that the authors have destroyed (or even significantly damaged) the DNA binding interface in the HTH region; it means that they have destroyed a critical specificity determinant. Therefore although it is consistent with the idea of another DNA binding interface, it is equally consistent with a damaged recognition (but not other contacts) of the HTH interface. This is particularly relevant since, in several ParB structures, there are several nonspecific, phosphate contacts with the DNA by the HTH. The authors are careful with their terminology, but it still seems somewhat misleading to not point out that the results are consistent with both possibilities.

5) "a secondary role for the CDBD in binding to nsDNA, albeit weakly compared to the *parS* sequence" – however not weakly compared to the CTD nsDNA binding activity. Given that only one ParB dimer can occupy *parS*, most ParBs presumably will bind nsDNA via the HTH in the context of condensed complexes/foci. This would still be important for the condensation activities. I would delete the word "secondary".

---

## [Author Response]

Essential revisions:1) The CTD-CTD interaction:The authors use the CTD peptide both in in vitro competition and in vivo interference assays. They observe an inhibition of DNA condensation in vitro, and the absence of foci formation in vivo, from which they conclude on the direct involvement of the CTD in these activities. However, another interpretation of the results would be that the CTD from the peptide and the full-length exchange between dimers resulting in equilibrium with heterodimers as the major species. In particular, in vivo this would lead to an important decrease in ParB2 concentration that would greatly reduce the possibility to spread on the DNA and thus to form visible foci. Moreover, the overexpression of CTD reduces significantly the specific binding of ParB to parS, while the spreading activity in the vicinity of parS is not reduced (Figure 5). This data appears in direct contradiction with the author's proposition of the roles of the CTD for which we would expect no reduction in parS binding and the absence of spreading away from parS. In the same vein, in vitro, the formation of heterodimers would also prevent the formation of the DNA binding domain (alignment of the two HTH motifs in the ParB dimer) resulting in the absence of DNA compaction.A major concern surrounds the implication that the C-terminal dimerization domain exists as a monomer (e.g. Figure 1) and/or can exchange during DNA binding (exchange is mentioned several times in the Results but not in the Discussion). As far as I know, in all ParBs tested at physiological, solution concentrations (low μm and nM), C-terminal fragments are always dimers and N-terminal fragments are never dimers. The N-terminal interactions have been uncovered only at crystallographic concentrations. In the third paragraph of the Introduction, for example, the compact dimer of T. thermophilus Spo0J referred to here is not seen in biochemical experiments in the same study – the N-terminal fragments are monomers in solution.In contrast to the situation proposed, the P1 ParB C-terminal dimer interaction is very strong and requires denaturation with GuHCl and renaturation to allow exchange and heterodimerization (JBC 2000, 275:8213). Otherwise dimer exchange does not happen during the time scale of DNA binding (> many minutes), although may occur with longer incubations (and likely would if coexpressed in vivo). The NMR H-D experiment cited but not presented (subsection “The structure of BsParB CTD reveals a dimer with a putative DNA binding interface”) does not allow interpretation without knowing concentrations and time scales. If exchange is significantly slower than DNA binding, then the ability to exchange is not relevant to the model. Similarly, the competition experiments in Figure 6 seem to preincubate the CTD with ParB, so it is not clear for how much time exchange has been allowed to occur. I would expect exchange to occur eventually if the two proteins are mixed together in vitro or expressed together in vivo, but the question of time-scale is critical here. Finally, exchange is not necessary for the bridging interactions to be necessary for condensation, nor do the data exclude roles for the N-terminal region (which previous studies, e.g. Graham et al., 2014, also show). I think the authors need to support the idea of exchange (during DNA binding) by reporting kinetics. They could measure the kD of dimerization directly. They could show that with their ParB, heterodimer ParB/DNA complexes form readily by EMSA, for example. Or their model could be modified or described differently if I have misinterpreted the implications here.

We showed that the truncated ParB-CTD construct was not only devoid of condensation activity itself, but also that it interfered with the formation of ParB networks by wild-type ParB both in vitro and in vivo. The referees were concerned about the molecular interpretation of this dominant negative effect.

In the manuscript, we discussed two non-exclusive explanations for the effect in vitro (see Figure 1; we will return later to the in vivo situation below). (1) The CTD simply competes with wild type ParB for binding to the DNA or (2/3) the CTD oligomerisation interface exchanges between the truncated CTD-only construct and the full length ParB. (Other formal explanations are possible although we think highly unlikely, e.g. the CTD-only construct acting as an unfoldase for the full length ParB.)

Explanation (1) can be excluded because a mutant CTD-only construct that can’t bind to DNA has the same effect. In scenarios (2) and (3), the CTD-only construct somehow poisons the wild type protein by forming heterodimers or heteroligomers that are unable to condense the DNA, and this could be thought about as occurring in two ways; either (2) heterodimerisation of the free ParB in solution inactivating the protein such that it no longer exchanges with the ParB network, eventually leading to decondensation or (3) heteroligomerisation on the ParB:DNA nucleoprotein complex that is being monitored in the magnetic tweezers (MT) apparatus, with the CTD-only construct acting as a “cap” for the CTD protein:protein interfaces, thereby dissolving the bridges in the network and leading to decondensation (see Figure 1 for cartoon). We *think* that it is the distinction between options (2) and (3) where the referees are concerned, and where we may have failed to fully discuss and/or explain the relevance of our data in discriminating between the two. Note that, in either case, we would argue that the dominant negative effect of the CTD points to a critical role of the CTD interface in ParB function, which is our central conclusion.

Alongside the original data, our new data suggest that option (2) is unlikely and provides direct evidence in support of (3). In line with the reviewers’ comments, and certainly in the case of the truncated CTD-only construct, the CTD interface does not exchange in free solution (i.e. in the *DNA-free state*) on a timescale that could simply titrate out the wild type protein dimer. Indeed, although our original NMR data shows that subunit exchange can occur, it is on a slower timescale (1/2 life >4hrs) than the effect we observe in minutes in the tweezers (see Figure 2 for details and discussion). On this basis, scenario (2) is unlikely.

However, the situation with the ParB nucleoprotein complex held in the MT apparatus is very different. The ParB binds co-operatively to DNA and transitions to a new, higher order configuration which leads to the DNA forming a compact condensed state under zero externally-applied force conditions. This condensed state is fully reversible by application of high force, which can be imposed by the MT apparatus (or potentially by the crowded nucleoid in vivo). By definition, these reversible force-dependent molecular interactions are the “bridges” that join DNA segments to condense the substrate. The dissociation rate for a non-covalent bond rises exponentially with opposing force, such that small changes in force can exert large effects on the exchange of the bridging interfaces (Bell, 1978; Evans and Ritchie, 1997), leaving them exposed for heterotypic interactions with appropriate interfaces in free solution. Under conditions where the CTD-only construct is at high concentration and in large excess over the ParB-DNA complex as in our MT experiments, it should rapidly interact with and “cap” the interface. This phenomenon is perhaps best demonstrated by the fact that one can repeatedly condense and decondense ParB-DNA nucleoprotein complexes by cycling through high and low force regimes, even in the absence of free ParB (Figure S5 in Taylor et al., 2015; original manuscript Figure 6Bi), but that this cycle is broken by the addition of free CTD to the system, leaving the DNA in a permanently decondensed state (original manuscript Figure 6bii).

Directly demonstrating that this subunit exchange occurs *on the DNA* is not trivial. In addition to performing blue-native PAGE on the free proteins, we have performed EMSA as suggested by the referees (Figure 2). The results are equivocal because the higher order complexes formed by ParB run in the wells, so we have no resolution to monitor the exchange of subunits. They do however suggest that the CTD does not prevent full length ParB from binding to DNA, which is a feature of model (3). Instead, we have recently developed a more direct method to test model (3), which uses a combined TIRF-MT correlative approach to monitor DNA condensation and fluorescent ParB/CTD binding simultaneously, under conditions of controlled external force (Figure 3). The experiments show that ParB binds to DNA without DNA condensation if the force is non-permissive (Figure 3), and that the ParB CTD inhibits the condensation of DNA-bound ParB without detectably displacing ParB from DNA (Figure 3). This shows that the inhibitory effect of the CTD on the full length ParB arises from the uncoupling of DNA binding from condensation, which strongly suggests to us that the CTD plays a role in bridging between segments of DNA in the network. Further experiments of this type will be the subject of a future publication.

We now return to the question of the dominant negative effect of the CTD in vivo, which is perhaps more straightforward to understand. As noted by the referees, since the polypeptides are translated together in the cell, they could heterodimerize immediately to form species that are inactivated and cannot go on to form higher order ParB structures (i.e. there is no need to invoke exchange of subunits). Alternatively, they could interfere with the ParB structures once formed as per our interpretation of the in vitro experiments described above. Again, we would argue that either scenario confirms the general importance of the CTD interface in the overall function of ParB in vivo.

Regarding Figure 7, the referees also stated that “the overexpression of CTD reduces significantly the specific binding of ParB to *parS*, while the spreading activity in the vicinity of *parS* is not reduced”. This would contradict our model, which predicts that, upon CTD overexpression, binding around *parS,* but not*precisely* at the *parS* sequence, would be reduced. Unfortunately, the ChIP-PCR method does not offer the resolution to measure specific binding alone, because regions of the DNA amplified very close to the *parS* sequence score for both specific binding and non-specific binding in the immediate vicinity. The data shows significant decreases in enrichment of DNA very close to *parS*, but also for other regions of non-specific DNA extending to ~3.5 kbp away from *parS* (original manuscript Figure 7; 359.45^o^). Note also that the background is higher for the CTD expression experiment, as defined by the percentage of DNA immunoprecipitated at the terminus (146^o^), such that the fold-enrichment at each of the probed sites is smaller than may be immediately apparent. We can conclude that non-specific DNA interactions are reduced, but we are unable to say whether specific interactions are also reduced, or are maintained at wild type levels.

Taking into account all of the above, we have made the following modifications to the manuscript:

We have added further details for the CTD-mediated decondensation experiment in the Materials and methods so that the timescale for potential exchange of substrates is clear to the reader. We have also clarified the timescale for HD exchange in the NMR experiments in the Results section.

We have produced a new figure (Figure 6—figure supplement 2 in the modified manuscript) which more thoroughly presents the possible mechanisms for ParB network formation. This figure also enables significant further discussion of ParB exchange, and the extent to which we can differentiate between different models for the dominant negative behaviour of the CTD in vitro and in vivo, including the lack of evidence for *rapid* exchange of ParB in free solution.

We have more explicitly explained the limitations in interpreting the ChIP-PCR experiment with respect to specific DNA binding.

2) The non-specific DNA binding activity:- In the absence of a totally defective mutant in the HTH motif, the author could not discriminate between the contribution of the HTH and the CTD domains for non-specific DNA binding. They cannot formally exclude that the HTH is not providing the main non-specific DNA binding activity as observed in some plasmid counterparts. Also, with the ParB^R149G^ variant, they cannot conclude that "the HTH motif is dispensable for non-specific binding and condensation" as point mutants in the second helix of HTH usually remains fully proficient for non-specific DNA binding.- The non-specific activity measured by EMSA corresponds to DNA that is stuck in the wells. This retention occurs at the same protein concentration for all tested variants (Figure 2). For the triple KKK mutant, which is proposed to be defective in ns-DNA binding activity, the same retention in the well is observed (Figure 4—figure supplement 1) and occurs at the same concentration as for the WT, thus preventing to conclude initially that this activity (retention in the wells) arises from the non-specific DNA binding activity of the CTD.- in vivo, the triple KKK mutant does not form foci as the WT ParB does. However, this variant displays a specific DNA binding activity on parS DNA that is strongly reduced (> 20-fold; compare Figure 4 to Figure 2), preventing directly the nucleation at parS and subsequently the formation of the foci. This variant could thus not serve as validating in vivo that the loss of the positively charge residues in the CTD prevents forming the ParB foci.- The authors use in their rational that ParB of plasmid P1 harbors a DNA binding activity in the CTD but this is due to an extra loop in this dimerization domain that binds specifically to direct repeats in the cognate parS sequence, and thus could not be compared per se to the putative non-specific DNA binding interface.

We show in this paper that the CTD of ParB has a non-specific DNA binding activity. The referees were concerned that we should not dismiss the role that the HtH motif may also play in this activity, partly by analogy to plasmid-encoded ParBs. We agree with this point, and had not intended to give the impression that the HtH motif and the CTD were exclusively responsible for specific and non-specific binding respectively. Indeed, several aspects of our own data highlight the fact that non-specific binding to DNA is more complex than a simple interaction of a CTD dimer with a duplex, and that there must be some overlapping function for the CDBD and CTD, especially in the context of condensed DNA (illustrated for example by the very interesting behaviour of the “KKK” mutant under conditions of high free [Mg^2+^] which the referees have highlighted).

The referees also questioned whether the KKK mutant could unequivocally establish the importance of the CTD nsDNA binding in vivo because, although nsDNA binding is completely abrogated, there is also defect in *parS* binding. We accept this point and have modified the Discussion accordingly (see below).

Finally, the referees pointed out that the role of the CTD of P1 is rather different to that proposed here for genomic BsParB. We accept this and had only intended to flag this up as a precedent for a DNA binding role in the CTD. Indeed, the binding mode and stoichiometry looks like it will be rather different. In the light of referees’ comments we have made the following modifications:

In three places in the manuscript we have changed "the HTH motif is dispensable for non-specific binding and condensation" (or similar) to "the R149 residue within the HTH motif is not required for non-specific binding and condensation" (or similar).

We now explicitly refer to the possibility that the CDBD/HTH region of the protein is also involved in nsDNA binding, with reference to the observation that large nucleoprotein complexes can still form on nsDNA even when the ability of the CTD to bind nsDNA has been effectively eliminated. New Figure 6—figure supplement 2 in the modified manuscript also illustrates this binding mode.

We have modified the Results and Discussion sections to acknowledge the *parS* binding deficiency of the KKK mutant and the implications of this for interpretation of the in vivo data.

[Editors' note: further revisions were requested prior to acceptance, as described below.]

The manuscript has been improved but there are some remaining issues that need to be addressed before acceptance. The reviewers comment on the need for clarification and rephrasing of interpretations in a number of places of the manuscript. While these issues can likely be addressed without the need for new experiments, a revised manuscript will need to include the suggestions put forward by the reviewers.Reviewer #2:The manuscript has been improved by adding schematics for the binding interfaces and by expending the discussion on the networking interactions and on the free-CTD experiments (Discussion, third paragraph). Also the authors have removed some over-interpretations as mentioned in their rebuttal. Lastly, some rewriting – for instance subsection “The CTD binds DNA non-specifically via a lysine-rich surface”, last paragraph; subsection “The CTD is critical for the formation of ParB foci in vivo”, first and third paragraphs – clarify the interpretation of the experiments.Amongst the new experiments provided within the rebuttal, it seems to me that at least Figure 2Ai-ii and Figure 2Bii should be included in the manuscript as supplementary data to support the arguing on the low exchange rate between subunits (as fully discussed in the new version).However, the authors do not provide a direct evidence for the respective contribution of the HTH and CTD motifs to non-specific DNA binding activity, while some rationales/arguments have not been changed thus some sentences are still misleading to me:- "we hypothesised that ParB contains a second DNA binding locus for nsDNA that functions independently of the helix-turn-helix motif. This idea was attractive to us for several reasons". As acknowledged by the authors in their rebuttal, the binding mode is very different for ParB of P1 as it binds specifically to dedicated motifs of parS. But, the 3rd reason still suggests that the CTD of ParB P1 is a precedent for nsDNA binding despite this argument should be removed.

The P1 structure does nevertheless provide a precedent for the CTD binding to DNA which we feel is highly relevant to our work. We appreciate that this is not a nonspecific DNA binding activity and that the binding mode is different. To be absolutely clear to the reader, we have modified the text in the Introduction to explain this.

“Thirdly, the distantly-related ParB protein from plasmid P1 provides a precedent for a second DNA binding locus in a Type I centromere binding protein (albeit an additional specific DNA binding site), and highlights the CTD as the putative candidate region”.

- "This is in contrast to the P1 ParB system where the CTD can bind two 16-mers". This comparison should not be made as this compares two different activities (specific versus non-specific binding) by two different interfaces (loops versus positively charged faces).

This related to point (1) above. We now explicitly clarify that the DNA binding mode is significantly different.

“This is in contrast to the P1 ParB system where the CTD operates in a different binding mode, and can bind two 16-mers”.

- "Together, these data show that mutation of the HtH motif effectively eliminates the ability of ParB to interact specifically with its cognate parS site, while nsDNA binding and condensation is relatively unaffected. This is consistent with the idea that nsDNA binding may occur at a second DNA binding locus." The specificity of binding is totally lost with the point mutation (R149G), but nothing is shown concerning the remaining non-specific DNA binding activity from the HtH motif. Thus, testing only for the loss of specific DNA binding could not provide a direct argument for searching a "second binding locus" for non-specific DNA binding and should not be written as such as a rational.

Given that we would subsequently directly demonstrate non-specific DNA binding at the CTD, we were attempting to guide the reader through our paper. For clarification we have modified the text to acknowledge other possible interpretations of the R149G mutation on nsDNA binding. (see also our response to reviewer #3, point 4).

“This is consistent either with the R149G mutation exclusively affecting nucleobase-specific contacts in the ParB-parS complex, and/or with the idea that nsDNA binding may occur at a second DNA binding locus.”

– "A striking feature of the structure is a highly electropositive face of the dimer arising from several conserved Lys residues analogous to the plasmid-encoded SopB and P1 ParB proteins". The authors have to discuss at some point the difference with SopB for which, despite the presence of the high electropositive face in the CTD, no residual non-specific DNA binding activity was reported in a triple HtH mutant by contrast to a point mutant in the HtH (equivalent to ParB-R149G) that eliminates the parS DNA binding but not nsDNA binding (Ah-seng et al., 2009), both mutants harboring the highly electropositive face of the CTD.

This is a fair and important point. We have now added a comment pointing out the distinction between ParB and SopB and the appropriate reference (Ah-seng et al., 2009).

[In both *B. subtilis* ParB and P1 ParB, the Lys/Arg rich surface has been shown to bind to DNA using structural or biochemical techniques ((Schumacher and Funnell, 2005; Schumacher, Mansoor and Funnell, 2007) and this work), but experiments with SopB do not support the idea that it shares this activity (Ah-seng et al., 2009).]

Reviewer #3:In this revision, Fisher et al. have further clarified the details of the activities and structure of the ParB C-terminal dimerization domain (CTD) in DNA binding and condensation. With one exception (see #1), they have addressed my concerns here. In this study, the major new information is the identification of non-specific DNA binding activity of the CTD. In addition, they present the first structure of the CTD of a bacterial ParB, which turns out to be similar to those of two plasmid ParBs published previously. The authors use sophisticated biophysical approaches, such as magnetic tweezers, to examine the condensation (considered as bridging) activities of ParB and their dependence on the CTD, and these approaches are uncommon and an elegant way to dissect these complexes. The idea that the dimerization domain is involved in bridging and condensation is not new and has been proposed and observed in several studies (e.g. Graham et al., 2014; Sanchez et al., 2015; Schumacher and Funnell, 2005; Schumacher, Mansoor and Funnell, 2007), but these experiments directly probe the role of the CTD in condensation.1) My major concern is with the models (represented in cartoons in Figure 1 and Figure 6—figure supplement 2, form Y), that the CTD is monomeric in solution, a concern from the original review, to which the authors did not respond here. In their figures, they represent ParB as dimerized either by CTD or NTD in solution (but not both). As mentioned in previous reviews, the dimer forms of Spo0J/ParB lacking the CTD were seen only in the crystal structures, and the T. thermophilus Spo0J fragment was shown in the same paper to be monomeric in solution. I also note that the DNA binding experiments in the latter study were also done at crystallographic, not physiological, concentrations of protein. As well as published experiments from other ParBs, the current authors' own experiments presented here show that the CTD exchanges extremely slowly in solution, evidence of a very tight kD. I accept their arguments here that exchange may be different when bound to DNA, but this is not represented in their cartoons. Similarly, the poor folding and insolubility of the CTD leucine mutants (dimer interface) also supports that the CTD is never "free".

We are not aware of any data reporting the protein:protein interfaces which are associated in free solution in full length genomic ParB. However, the referee is correct to point out that we failed to illustrate the possibility that both CTD and NTD interfaces are satisfied in free solution. We have now illustrated all possible dimer structures in our cartoon (Figure 6—figure supplement 2) and have also indicated in the legend that scenarios in which the CTD is dimerised may be considered to be more likely because the purified CTD is a tight dimer in free solution.

“In free solution, the interfaces supporting dimerization of ParB are unclear, but scenarios in which the CTD is dimerised may be considered to be more likely because the purified CTD has been shown to be a tight dimer.”

This (CTD monomers in solution) may not be what the authors intend, but it is misleading and needs to be changed (or clarified) in these cartoons.2) Are the CTD or ParB KK and KKK mutants still dimers? The NMR structure clearly predicts that these mutations do not damage the dimer interface, and I expect this to be the case, but this should be confirmed experimentally and reported. Otherwise the authors cannot conclude that the effects of these mutations can be directly attributed to DNA binding defects. Similarly, they argue that CTD interference is due to dimerization since the KKK mutants also interfere; this is a circular argument without independent evidence that the mutations do not interfere with dimerization.

Yes. We showed in the paper that WT, KK and KKK variants of the CTD display similar CD spectra (Figure 3—figure supplement 1 and Figure 4—figure supplement 1). We now also explain in the Materials and methods that the WT, KK and KKK variants of the CTD (and the full length protein) run almost identically on preparative size exclusion columns (Author response image 1 shows normalised UV traces for elution of the CTDs indicated from a Superdex 75 column).

“The wild type, K255A+K257A and K252A+K255A+K259A variants of ParB behaved equivalently during purification, and run almost identically on preparative size exclusion columns (both in the context of the CTD and full length protein), suggesting that they are all dimeric.”

3) "we showed that the CTD is not required for parS binding": where are the experiments in this study to support the conclusion; i.e. parS binding in the absence of the CTD? The ParB/CTD-KKK mutants presumably still dimerize (comment #2) and indeed they contribute to the affinity of parS binding. I think that they mean that the CTD DNA binding interface is not required for parS binding, which should be clarified.

The referee is quite correct here; we are referring to the binding interface and not the entire CTD domain. We have corrected this in the text.

“We showed that the DNA binding interface in the CTD is not required for parS binding, and that this is instead dependent on the HtH motif found within the CDBD domain as predicted in several previous studies.”

4) Subsection “The R149 residue within the HTH motif is essential for specific binding to parS, but not required for non-specific binding and condensation”, last sentence: The lack of specific DNA binding by the R149G mutation does not mean that the authors have destroyed (or even significantly damaged) the DNA binding interface in the HTH region; it means that they have destroyed a critical specificity determinant. Therefore although it is consistent with the idea of another DNA binding interface, it is equally consistent with a damaged recognition (but not other contacts) of the HTH interface. This is particularly relevant since, in several ParB structures, there are several nonspecific, phosphate contacts with the DNA by the HTH. The authors are careful with their terminology, but it still seems somewhat misleading to not point out that the results are consistent with both possibilities.

This is related to point (3) made by the reviewer #2 above. The text now explains that the lack of specific binding in R149G is consistent with two possible interpretations that are not mutually exclusive. See above.

5) "a secondary role for the CDBD in binding to nsDNA, albeit weakly compared to the parS sequence" – however not weakly compared to the CTD nsDNA binding activity. Given that only one ParB dimer can occupy parS, most ParBs presumably will bind nsDNA via the HTH in the context of condensed complexes/foci. This would still be important for the condensation activities. I would delete the word "secondary".

We accept this point and have changed the text accordingly.